# Droughts worsen air quality and health by shifting power generation

Mathilda Eriksson [1,3] ✉, Alejandro del Valle [1,3] ✉ & Alejandro de la Fuente[2]

Fine particulate matter ($PM_{2.5}$) is a leading environmental cause of mortality. Droughts can worsen air quality in regions that rely on hydropower by shifting energy production to combustion power plants. This study quantifies drought-induced excess $PM_{2.5}$ in Latin America and the Caribbean, where over 443 million people live within 50 km of a combustion power plant. Leveraging a monthly plant-level panel spanning 20 years, we link hydrological droughts, measured as negative runoff anomalies in hydropower watersheds, to changes in $PM_{2.5}$ concentrations near combustion power plants. Our analysis reveals that these droughts lead to an average increase of 0.83 μg m$^{-3}$ in $PM_{2.5}$ levels. Counterfactual simulations for the region reveal that this excess $PM_{2.5}$ results in up to 10,000 premature deaths annually. Combining our estimates with climate, demographic, and combustion power plant phase-out projections, we demonstrate that this health burden will persist over the next four decades without targeted interventions.

Electricity generation is a water-intensive process, with most power plants requiring water to spin hydroelectric turbines or cool thermo-electric generators[1]. In Latin America and the Caribbean (LAC), approximately half of total electricity generation comes from hydro-power, while the other half is derived from combustion power plants fueled by coal, oil, gas, and biomass[2]. Because droughts predominantly limit the generation capacity of hydropower plants, droughts can shift generation to combustion power plants. A plausibly important but understudied consequence of this shift in generation is the worsening of local air quality. Among the pollutants released during combustion, fine particulate matter ($PM_{2.5}$, particles smaller than 2.5 μm) warrants particular attention for its ability to disperse over wide areas[3–7], and its well-documented adverse health effects[8–12], even at low concentrations[13]. With over 443 million individuals residing near combustion power plants and climate change anticipated to exacerbate the frequency and severity of droughts in the region, quantifying the extent of drought-induced excess $PM_{2.5}$ is a critical empirical question with profound implications for public health and energy policy.

To address this question, we assemble a monthly frequency power plant-level panel covering the 2000–2020 period. The panel provides information on $PM_{2.5}$ concentrations in the proximity of combustion power plants and market-level measures of hydrological drought affecting hydropower plants. We focus on hydrological droughts as they directly reflect water availability in rivers and reservoirs. To measure hydrological drought, we use runoff anomalies, defined as deviations from long-term mean runoff levels, calculated within the watersheds that supply hydropower plants. These anomalies provide a granular and localized indicator of drought conditions affecting water resources essential for energy production. To capture broader market-level impacts, we aggregate these watershed-level runoff anomalies using multiple methods, ensuring that our measures accurately represent the overall degree of hydrological drought faced by hydropower plants while preserving their localized relevance to generation capacity. Our preferred market-level measure is the fraction of hydropower generation capacity affected by drought (FHD). The assembled panel also provides information on an extensive set of meteorological factors, wildfire emissions, proxies of electricity demand, and characteristics of power plants. Using this dataset, we estimate the excess $PM_{2.5}$ generated by hydrological droughts. Our empirical strategy combines fixed effects methods, which allow us to control for unobserved time-invariant characteristics, common

[1]Maurice R. Greenberg School of Risk Sciences, Georgia State University, Atlanta, GA, USA. [2]Poverty and Equity Global Practice, World Bank Group, Delta Center, Upper Hill, Nairobi, Kenya. [3]These authors contributed equally: Mathilda Eriksson, Alejandro del Valle. ✉e-mail: meriksson@gsu.edu; adelvalle@gsu.edu

time-varying shocks, and market-specific seasonal variations, with post-double selection methods, which enable us to flexibly control for observed confounders. Our estimates have a causal interpretation because, conditional on meteorological factors, changes in electricity demand, and the extensive set of fixed effects, hydrological droughts create a plausibly exogenous shock to hydropower generation.

To distinguish the impact of drought-induced shifts in energy generation from alternative mechanisms, such as wildfire emissions, we implement several methodological safeguards. In our main analysis, to minimize wildfire influence we exclude plant-month observations where wildfire emissions are detected within a 50 km radius. Additional robustness checks extend the exclusion radii and remove observations potentially affected by dust storms, addressing another significant source of air quality degradation associated with hydrological droughts. Placebo tests further reinforce our effort to pinpoint the mechanism, as they reveal no relationship between hydrological droughts and PM$_{2.5}$ near non-combustion power plants or during periods before combustion power plants became operational. Additionally, we analyze the heterogeneous effects of droughts with respect to plant characteristics and uncover patterns that strongly support the conclusion that hydrological droughts drive shifts in energy generation, leading to increased PM$_{2.5}$ levels.

In this work, we demonstrate through multiple lines of evidence, including robust controls, extensive exclusion analyses, placebo tests, and observed patterns of heterogeneous effects, that hydrological droughts pose a significant public health risk as they considerably increase PM$_{2.5}$ concentrations through their impact on energy generation. Our paper contributes to several strands of literature. Most directly, it provides causal evidence of how hydrological droughts affect air quality through shifts in energy generation in middle-income countries, extending recent work focused on the western United States[14–16]. Our findings show that drought-induced changes in energy generation are an important driver of PM$_{2.5}$ across LAC. By examining this relationship in a region where energy systems, exposure patterns, and environmental conditions differ markedly from those in the U.S., we uncover three insights into how drought-induced shifts in energy generation impact air quality in middle-income economies. First, we demonstrate that even short-duration droughts (≤3 months) can substantially impact air quality, likely due to the vulnerability of the region's run-of-river hydropower infrastructure, which depends heavily on consistent water flow and lacks substantial storage capacity. Second, we identify small-capacity combustion power plants, which are typically air-cooled and serve as marginal energy sources in the region, as significant contributors to excess PM$_{2.5}$ during drought periods. Third, we show that oil and biomass combustion power plants, which are widespread in LAC but rare in the U.S., emerge as major contributors to air quality degradation during droughts. Together, these findings expand our understanding of how droughts affect energy systems and air quality across diverse contexts, complementing evidence from the U.S. while underscoring the unique challenges faced by middle-income economies.

We also contribute to the environmental justice literature[17–20] by documenting that excess PM$_{2.5}$ falls disproportionately among those with lower socioeconomic status. This finding highlights a channel through which adaptation to droughts, by shifting generation to combustion power, can exacerbate existing inequalities. Our results have important implications for ensuring that energy transition policies and infrastructure planning consider distributive impacts, particularly as countries in LAC work to balance climate resilience with environmental justice goals.

More broadly, this study contributes to research quantifying the economic costs of droughts and climate change[21–26] by offering plant-level estimates of the costs associated with drought-induced excess PM$_{2.5}$, expressed in terms of premature deaths and associated monetized losses. Our analysis captures both accrued societal costs and projected losses through 2059, accounting for various climate-forcing scenarios and power plant phase-out policies. Importantly, our projections reveal significant regional variation in future hydrological drought trends, with most of LAC expected to experience an increase in droughts, while the Andean Region (Colombia, Ecuador, and Peru) is projected to see a decrease. Despite this regional heterogeneity, our projections indicate that, in the absence of policy action, the health burden of drought-induced excess PM$_{2.5}$ is likely to persist. By quantifying these previously unmeasured impacts, our work helps establish the full costs of droughts and provides critical evidence to evaluate both adaptation strategies and mitigation policies. These granular cost estimates are particularly valuable for policymakers weighing investments in energy storage, regional grid interconnection, and other infrastructure solutions that could reduce reliance on combustion power plants as marginal energy sources during droughts.

## Results

### Droughts affecting hydropower lead to excess PM$_{2.5}$

Our analysis reveals that hydrological droughts significantly increase PM$_{2.5}$ concentrations near combustion power plants in LAC, even when considering short-duration droughts. We examine the impact of droughts using the fraction of hydropower generation capacity affected by drought (FHD), which measures the proportion of hydropower capacity experiencing below-normal water conditions over a three-month window. As briefly described in the introduction, our primary analysis employs a linear regression model (Eq. (1)), hereafter referred to as the benchmark model, to quantify the relationship between FHD and PM$_{2.5}$ concentrations while controlling for various meteorological factors, proxies of electricity demand, and an extensive set of fixed effects. To ensure the robustness of our findings, we conduct a comprehensive series of sensitivity analyses, detailed in the Methods section.

We find that when all hydropower generation experiences drought (FHD = 1), PM$_{2.5}$ concentrations within 50 km of combustion power plants increase by 1.55 μg m$^{-3}$ (p-value = 0.000) compared to non-drought conditions (Fig. 1a). This substantial impact of short-duration droughts (3 months) highlights the vulnerability of the region's hydropower infrastructure to even brief periods of hydrological drought. We observe consistent results when analyzing alternative drought durations (see Supplementary Methods 1).

Figure 1b contextualizes the previous result by plotting the implied total PM$_{2.5}$ concentrations across different levels of FHD. The benchmark model predicts baseline concentrations of 15.76 μg m$^{-3}$ without drought, already well above the WHO guideline of 5 μg m$^{-3}$ for harmful concentrations[27]. These levels rise to 16.59 μg m$^{-3}$ at mean FHD and reach 17.31 μg m$^{-3}$ at maximum observed FHD. The drought-induced increase of 0.83 μg m$^{-3}$ at mean FHD levels represents a significant additional health risk in areas already burdened by high PM$_{2.5}$ levels.

To further investigate the relationship between drought severity and PM$_{2.5}$, we employ a dose-response model (Fig. 1c). Consistent with our hypothesis that hydrological droughts increase pollution by shifting energy generation to combustion power plants, we find a monotonic increase in PM$_{2.5}$ concentrations as the FHD rises. Notably, the impact on PM$_{2.5}$ is substantially larger when at least half of hydropower generation is affected by drought. Given that the dose-response model yields comparable results to those of the more parsimonious benchmark model (Fig. 1b, d), we focus on the benchmark model for the remainder of the analysis.

### Shifts to combustion power explain excess PM$_{2.5}$

Having established that hydrological droughts lead to excess PM$_{2.5}$ concentrations, we investigate alternative mechanisms that could explain our results. Most prominently, droughts increase the likelihood of wildfires, one of the primary sources of PM$_{2.5}$ pollution[28–31], as

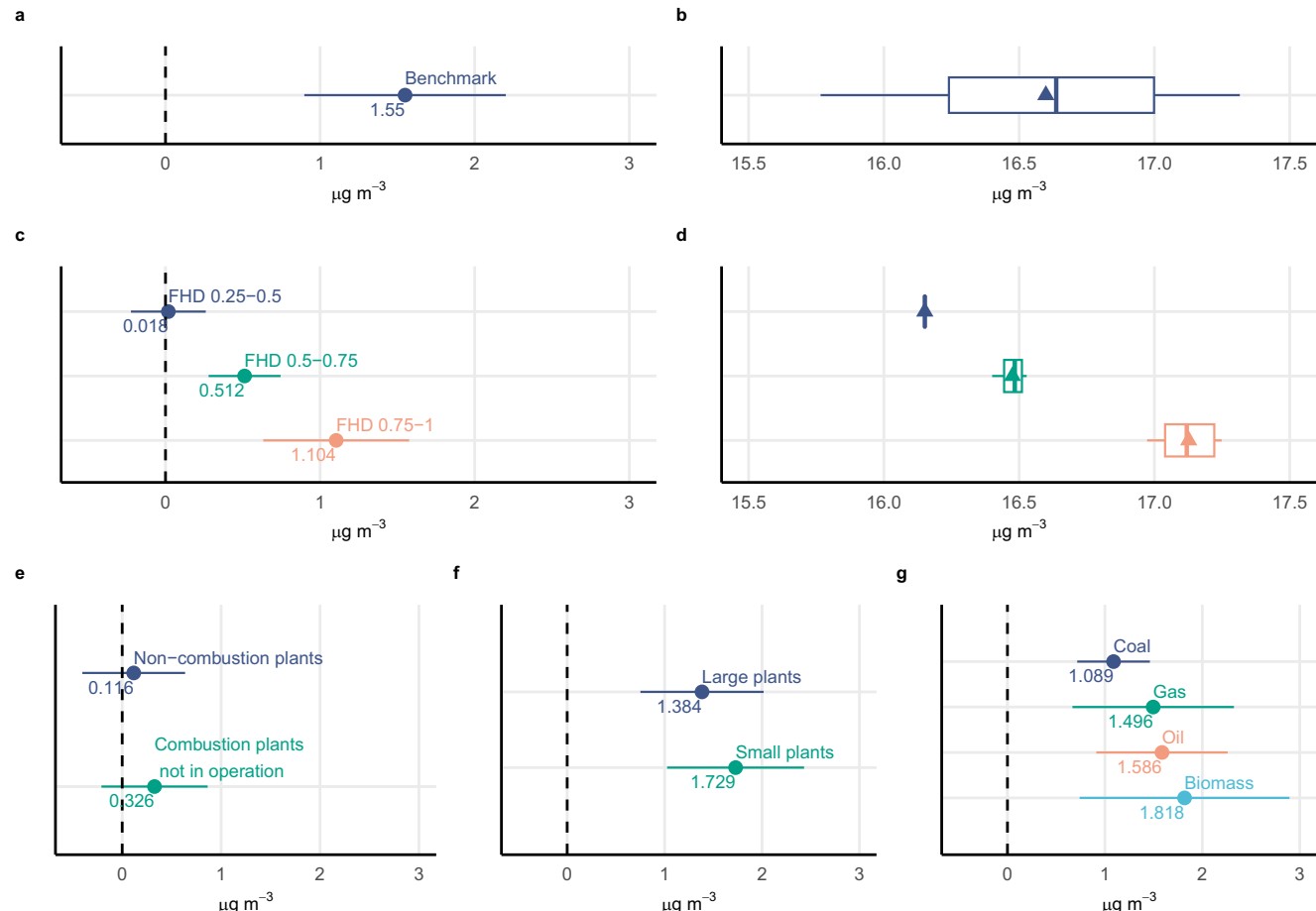

**Fig. 1 | Effect of the fraction of hydropower generation affected by drought (FHD) on PM$_{2.5}$ concentration, dose-response model, placebo tests, and heterogeneity with respect to size and fuel type.** All results are based on the analysis sample ($N$ = 79,022 plant-month observations), unless otherwise stated. **a** Point shows the $\beta$ coefficient from Eq. (1), with error bars indicating 95% confidence intervals (CI) derived from standard errors clustered at the market level (19 clusters). **b** Distribution of the implied total PM$_{2.5}$ concentrations, i.e., the marginal effect of the coefficient presented in (**a**) plus the predicted level of PM$_{2.5}$ in the absence of droughts. **c** Regression coefficients (points) and 95% CI (error bars) from a dose-response model where the FHD variable is discretized in four groups. The reference group is FHD less than 0.25. **d** Distribution of the implied total PM$_{2.5}$ concentrations, i.e., the marginal effect of each coefficient presented in (**c**) plus the predicted level of PM$_{2.5}$ for the reference group. **b**, **d** Box plots indicate median (middle line), 25th, 75th percentile (box), and minimum and maximum (whiskers) as well as mean values (triangles). **e** Regression coefficients (points) and 95% CI (error bars) from two placebo exercises, i.e., the impact of FHD on PM$_{2.5}$ around non-combustion power plants ($N$ = 62,066 plant-month observations) and around combustion power plants before they are operational. Non-combustion power plants include wind, solar, geothermal, and nuclear. The placebos exclude plants with combustion power plants operating within a 50 km radius. **f**, **g** Regression coefficients (points) and 95% CI (error bars) of the impact of FHD on PM$_{2.5}$ for each size and fuel type sub-group. **f** Coefficient for large power plants (≥30 MW) is statistically different from small plants ($p$-value = 0.053). **g** Coefficient for coal plants is statistically different from oil plants ($p$-value = 0.038). Reported $p$-values correspond to two-sided $t$-tests without adjustment for multiple comparisons. Source data are provided as a Source Data file (sourcedata.xlsx). The data and code used to obtain the estimates are available at https://www.openicpsr.org/openicpsr/project/217201.

well as increase the occurrence of dust storms[32]. To address these potential confounding factors, our primary analysis excludes plant-month observations with fire emissions within a 50 km radius. As described in Supplementary Methods 1, our robustness checks demonstrate that our results are robust when extending the exclusion radius up to 100 km for both fire and dust emissions. While these restrictions reduce concerns about wildfires and dust emissions driving our findings, they do not entirely rule out their influence, as PM$_{2.5}$ can remain suspended in the atmosphere for several days and travel hundreds of kilometers.

To further disentangle the impact of droughts on PM$_{2.5}$ through a shift in generation to combustion power from other potential mechanisms, we conduct two placebo exercises (Fig. 1e). In the first placebo exercise, we compute PM$_{2.5}$ concentrations within a 50 km radius of non-combustion power plants (i.e., wind, solar, geothermal, and nuclear) and exclude observations with combustion power plants within that radius. We then estimate Eq. (1) using this sample. If

wildfires or dust storms drive the excess air pollution, we should also expect to observe an increase in air pollution in this sample of non-combustion power plants. However, consistent with the idea that wildfires and dust storms are not the primary mechanism for excess air pollution, we find that the impact of FHD on air pollution around non-combustion power plants is small and statistically indistinguishable from zero ($p$-value = 0.638).

One important caveat with the previous exercise is that the location of combustion and non-combustion power plants may differ systematically, and the resulting null effect may reflect the differences in geographic characteristics. To address this limitation, the second placebo exercise tests whether air pollution increased among combustion power plants in the years before they were operational. Once again, we cannot reject the null hypothesis that the impact of FHD on air pollution is different from zero ($p$-value = 0.194). Together, these placebo exercises indicate that wildfires and dust storms are unlikely to drive our results.

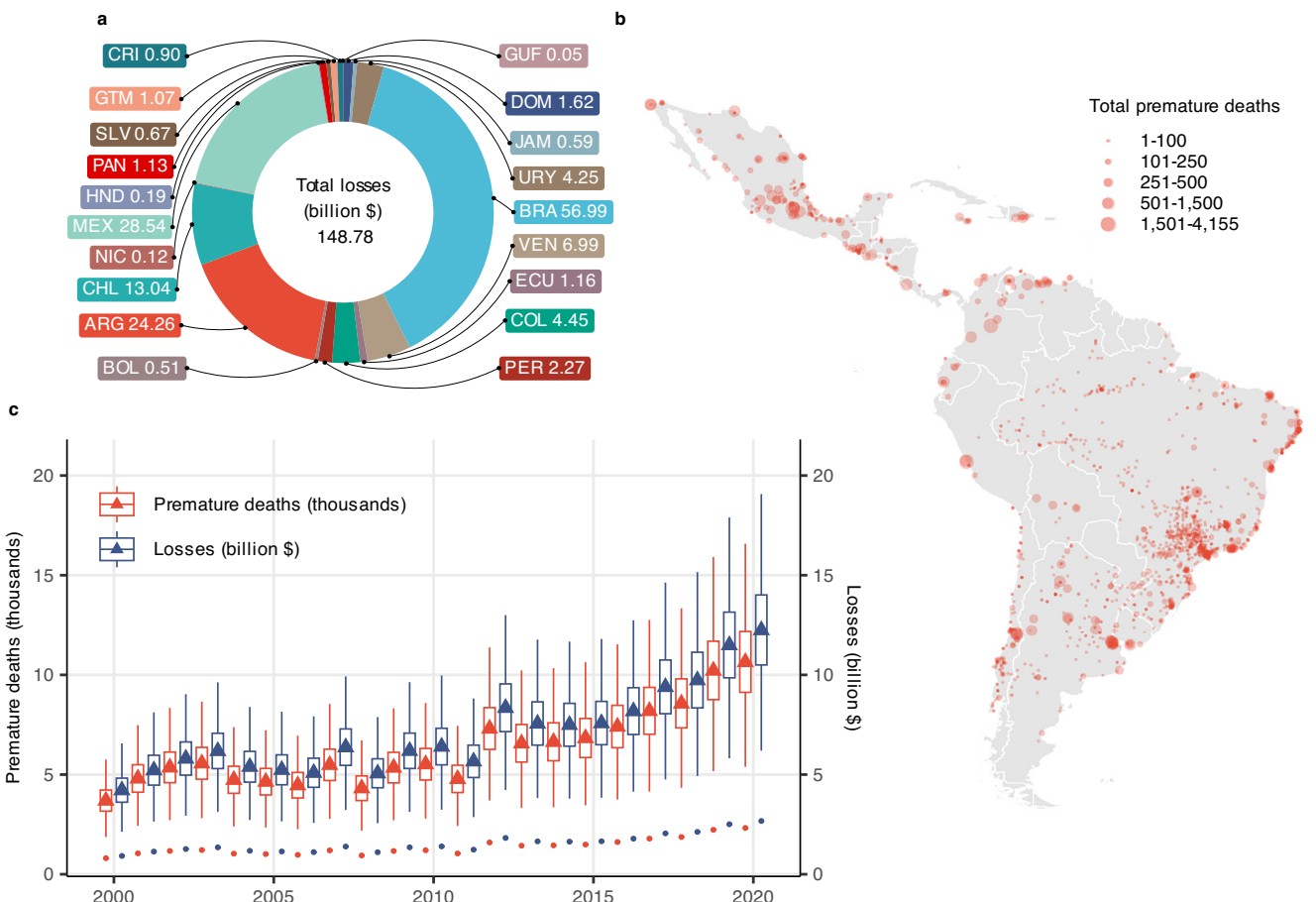

**Fig. 2 | Premature deaths and monetized losses from drought-induced excess PM₂.₅.** Premature deaths are calculated using the estimate in Fig. 1a and the concentration-response function of Deryugina et al.[35]. Premature deaths are monetized to 2019 USD using country-year estimates of the value of a statistical life extrapolated from US estimates. **a** Country-level distribution of cumulative losses between 2000 and 2020. Country codes are as follows: CRI (Costa Rica), GTM (Guatemala), MEX (Mexico), BRA (Brazil), SLV (El Salvador), PAN (Panama), HND (Honduras), NIC (Nicaragua), CHL (Chile), COL (Colombia), ARG (Argentina), PER (Peru), BOL (Bolivia), GUF (French Guiana), DOM (Dominican Republic), JAM (Jamaica), URY (Uruguay), VEN (Venezuela), and ECU (Ecuador). **b** Plant-level distribution of cumulative premature deaths between 2000 and 2020. **c** LAC-level

annual time series of premature deaths and losses ($N$ = 21,000 simulation-year units; 1000 draws × 21 years). The spread of the box plots displayed each year results from the uncertainty in our estimation of excess PM₂.₅ concentrations and the value of the observed FHD. Box plots indicate median (middle line), 25th, 75th percentile (box), 1.5 times the interquartile range (whiskers), outliers (single points), and mean values (triangles). Administrative boundary data were obtained from the Database of Global Administrative Areas (GADM), version 4.1, available at www.gadm.org. Source data are provided as a Source Data file (sourcedata.xlsx). The data and code used to obtain the estimates are available at https://www.openicpsr.org/openicpsr/project/217201.

To further investigate whether the estimated increase in PM₂.₅ during hydrological droughts is consistent with a shift in energy generation towards combustion power plants, we examine how the effect varies with plant size and fuel source. Figure 1f, g presents the drought-induced excess PM₂.₅ for each subgroup of combustion power plants. While hydrological droughts lead to increased air pollution across all subgroups, the magnitude of the effect is notably larger for smaller capacity plants (<30 MW) and those using oil or biomass as fuel.

This pattern of heterogeneous effects aligns with the energy shift mechanism in several ways. First, larger coal and gas power plants typically serve as baseload electricity generation, operating continuously near maximum capacity[1]. Consequently, these plants have limited spare capacity to increase output during droughts, resulting in a more muted pollution response compared to oil or biomass plants. Second, large baseload power plants tend to be more water-intensive than their smaller counterparts[1], further constraining their ability to ramp up production during water-scarce periods. In contrast, smaller combustion power plants, which in our sample primarily consist of biomass and oil plants, more frequently utilize air-cooling methods, allowing for greater flexibility in adjusting output. Notably, the most

substantial increase in PM₂.₅ occurs around biomass power plants, a well-known source of particulate pollution[33,34]. Taken together, the pattern of heterogeneous effects by plant size and fuel source provides strong supporting evidence for the hypothesis that our estimates of the impact of hydrological droughts on PM₂.₅ are primarily driven by shifts in electricity generation toward combustion power plants.

## Lives lost due to excess PM₂.₅

To assess the human health consequences of drought-induced air pollution, we estimate premature deaths using our estimates of excess PM₂.₅, counts of exposed population, and the causal concentration-response function (CRF) from Deryugina et al.[35]. Figure 2b presents the distribution of cumulative drought-induced premature deaths between 2000 and 2020, revealing that these deaths are widespread across the region. Nearly every country has at least one power plant linked to over 250 premature deaths within this period. Figure 2c plots the annual time series of drought-induced premature deaths aggregated at the LAC level. The spread of the box plot displayed each year results from the uncertainty in our estimation of excess PM₂.₅ concentrations and the observed FHD. The figure reveals that the mean

premature deaths per year (red triangle markers) range from 3692 to 10,641. Supplementary Fig. 1 extends the analysis to include alternative well-known CRFs[36–39], yielding mean annual premature deaths ranging roughly from 700 to 13,500. Notably, the estimates based on our preferred CRF[35] lie near the midpoint of this range, reflecting their alignment with the broader set of CRFs considered.

A potential limitation of the available CRFs is that they do not account for the possibly greater vulnerability to $PM_{2.5}$ among populations with lower socioeconomic status. To descriptively assess whether populations residing near power plants tend to have lower socioeconomic status compared to the overall population, we use downscaled Human Development Index (HDI) data[40] to compute the power plant-level mean HDI relative to country-level HDI. As shown in Supplementary Fig. 2, four out of five combustion power plants have nearby populations with significantly lower HDI levels. This pattern is consistent across all types of combustion power plants (i.e., coal, gas, oil, biomass) and aligns with existing literature, which highlights the negative association between socioeconomic status and $PM_{2.5}$ exposure[18–20,41]. Given the prevalent residential pattern of the population with lower socioeconomic status in LAC and the available CRFs, it is likely that our estimates underestimate the premature deaths caused by excess $PM_{2.5}$.

To quantify the economic value of these lives lost, we combine our premature death estimates with country and year-specific values of a statistical life (VSL). While we recognize that this metric does not fully capture the intrinsic value of human life or the lived experiences, it provides a useful standardized measure for policy evaluation. Figure 2 shows that cumulative losses for the region between 2000 and 2020 are in the order of \$150 billion (2019 USD), with Brazil, Mexico, and Argentina each experiencing cumulative losses of over \$24 billion. Figure 2c plots the annual time series of these drought-induced losses aggregated at the LAC level. The figure further reveals that yearly mean losses (blue triangle markers) are in the order of \$4.2–12.2 billion. An analogous exercise using using alternative CRFs[36–39] yields mean annual losses of 0.8–15.0 billion (Supplementary Fig. 3). Notably, these monetized longevity losses may underestimate the total social cost of worsened air quality triggered by droughts, as our counterfactual simulations do not account for decreased quality of life[42] or non-health effects, such as impacts on productivity and cognitive ability[43–46].

## Projected lives lost due to excess $PM_{2.5}$

To assess the future implications of our findings, we present projections of hydropower generation affected by drought and the associated health costs of drought-induced excess $PM_{2.5}$. Figure 3a illustrates the projected percentage change in the mean fraction of hydropower generation affected by droughts during 2020–2059, relative to the baseline period of 2000–2019. These projections are derived from runoff data provided by 22 climate and earth system models under three climate scenarios: SSP1–2.6, SSP2–4.5, and SSP3–7.0, which combine Shared Socio-economic Pathways (SSPs) and Representative Concentration Pathways (RCPs). These scenarios span a range of potential future climate forcing, with higher numbers indicating more severe climate change. The box plots represent the variability across model projections, while the triangle markers indicate the ensemble means. For the region, most models indicate an increase in the fraction of hydropower generation affected by hydrological droughts. The ensemble mean suggests that the FHD will likely increase between 22 and 24%. The ranking in the FHD increase is consistent with higher climate-forcing scenarios leading to a higher drought exposure. However, differences across scenarios are projected to remain small by 2059. Figure 3b–f presents results from analogous analysis for each IEA sub-region. Nearly all sub-regions are expected to experience an increase in FHD, with sub-regions like the Caribbean and Southern South America particularly affected.

The exception is the Andean Region (Colombia, Ecuador, and Peru), which is expected to see a reduction in the FHD of roughly 25%.

Building on the evidence that droughts will continue to affect hydropower generation across the region, we extend our analysis to project the evolution of premature deaths and the associated monetized losses caused by drought-induced excess $PM_{2.5}$. These projections integrate climate, demographic, and economic growth trends under the SSP-RCPs paired with IEA energy policy scenarios that outline potential retirement schedules for combustion power plants. The three energy policy scenarios, APS, STEPS, and RES, reflect progressively fewer retirements, ranging from full implementation of announced pledges to no retirements (see Methods for details).

Figure 3g shows observed mean premature deaths and 66 projected paths (3 scenarios × 22 models). To illustrate the overall trends, we include LOESS curve fits (bold lines) for each scenario. The figure reveals that there is substantial uncertainty in the projected number of premature deaths. Only under SSP1–2.6-APS do deaths stabilize at approximately 5000 per year by 2059, a level comparable to that observed in the early 2000s. In other scenarios, premature deaths increase significantly. For instance, under SSP3–7.0-RES, premature deaths rise to approximately 30,000 annually by 2059, nearly six times the levels observed in SSP1–2.6-APS. Figure 3h presents the monetized losses associated with these premature deaths. Consistent with the previous results, projected losses are expected to persist or increase relative to the early 2000s.

To explore the drivers of these projections, Supplementary Fig. 4 depicts the evolution of premature deaths and economic losses across SSP-RCP combinations for each energy policy scenario. The figure underscores the dominant influence of energy policy, with ambitious plant retirements under APS significantly reducing premature deaths and economic losses compared to STEPS or RES scenarios. Within each energy policy scenario, differences between SSP-RCPs depend on climate, demographic, and economic factors. The figure shows that the largest projected increase in deaths occurs under SSP1–2.6, driven by the rapid aging of the population, which offsets the relatively minor differences in runoff across SSP-RCP scenarios by 2059. This finding highlights the significant role demographic factors play in shaping exposure to this hazard over this time horizon. Additionally, the figure shows that economic losses are further amplified by higher economic growth, which increases the VSL. These findings highlight that climate mitigation measures alone are insufficient and that targeted policies, such as accelerating plant retirements and reducing population exposure, are crucial to addressing these losses.

## Discussion

Over the past century, droughts have intensified, extending their duration and geographic reach, with documented impacts on ecosystems, economic activity, and public health[47]. One understudied pathway through which droughts affect human health is by disrupting electricity generation. This study builds on previous evidence for this pathway from the western United States[14–16] and provides multi-country evidence linking hydrological droughts to worsening air quality and health through shifts in electricity generation from hydropower to combustion power plants.

The observed relationship reveals that droughts affecting hydropower watersheds lead to measurable increases in $PM_{2.5}$ concentrations near combustion power plants. Dose-response estimates, placebo tests, heterogeneity analyses, and a broad battery of robustness checks support the interpretation that drought-induced shifts in power generation drive these increases. The analysis reveals distinct vulnerability patterns in the region, marked by the susceptibility of hydropower infrastructure to even short-duration droughts and a heavy reliance on oil and biomass combustion plants as marginal sources of energy.

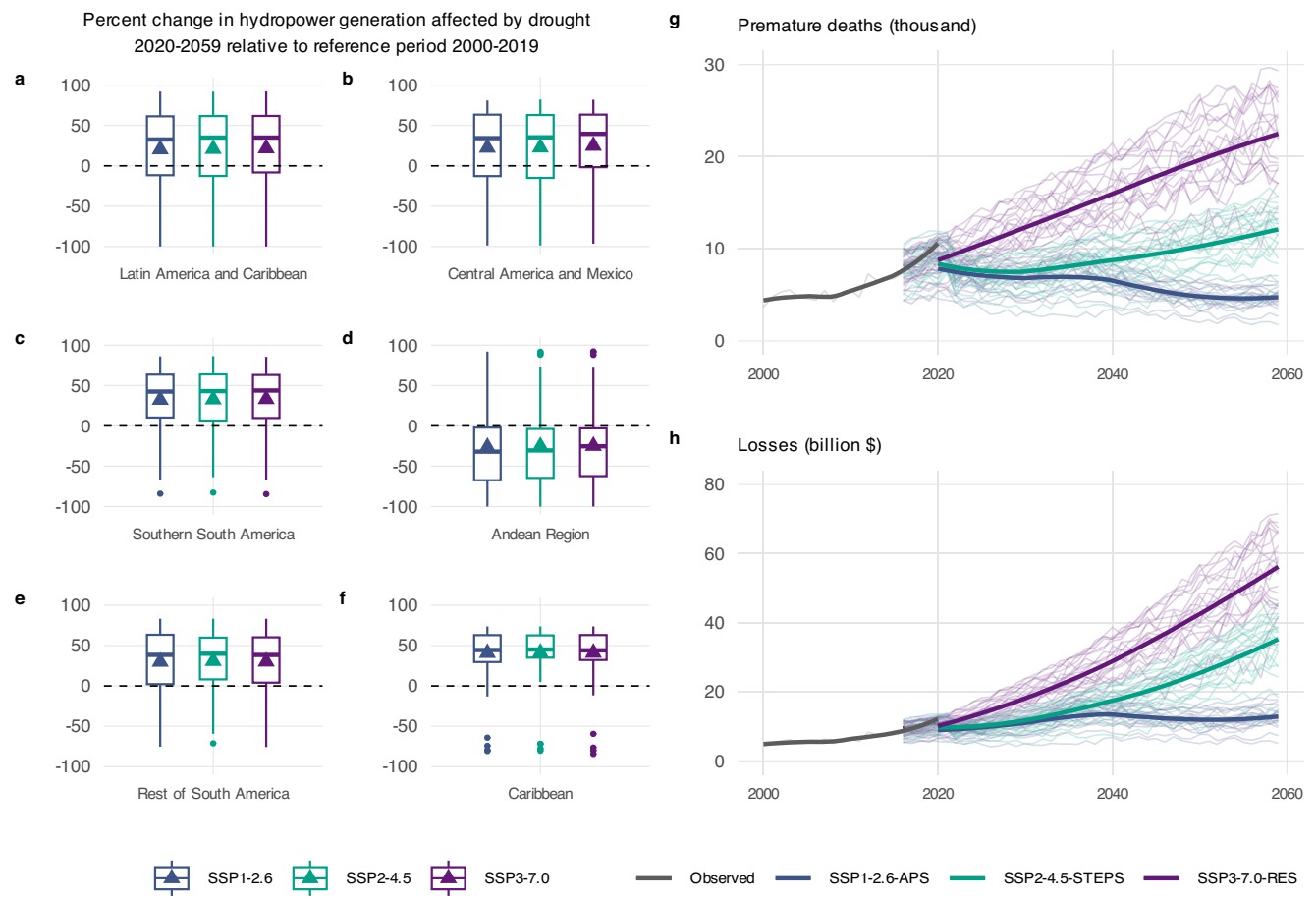

**Fig. 3 | Projected hydropower exposure to drought and losses from drought-induced excess PM$_{2.5}$. a–f** Percent change in the mean fraction of hydropower generation affected by drought 2020–2059 relative to the mean 2000–2019. Each box plot represents one of three Shared Socioeconomic Pathways (SSPs) paired with Representative Concentration Pathways (RCPs) (SSP-RCP scenarios). The distribution of runoff projections from the 22 climate and earth system models determines the box plot's spread. Box plots indicate median (middle line), 25th, 75th percentile (box), 1.5 times the interquartile range (whiskers), outliers (single points), and mean values (triangles). The IEA sub-regions are: **b** Central America and Mexico, **c** Southern South America (Argentina, Bolivia, and Chile), **d** Andean Region (Colombia, Ecuador, and Peru), **e** Rest of South America (Brazil, Venezuela, Paraguay, and Uruguay), and **f** Caribbean. **g**, **h** Plot the observed and projected evolution of premature deaths and the corresponding monetized losses between 2000

and 2059. Estimates of future losses are derived from demographic, climate, and economic projections under the three SSP-RCP scenarios. The retirement schedule of combustion power plants follows electricity sector scenarios from IEA[69]. The Announced Pledges Scenario (APS) assumes retirements in line with existing country pledges. The Stated Policies Scenario (STEPS) assumes retirements in line with the IEA's current assessment of the region's energy direction. The Reference Electricity Scenario (RES) assumes no retirements. All 66 projections (22 models × 3 scenarios) also assume that additional electricity demand is met with non-combustion power, that no new combustion power plants are introduced, and that the concentration-response function is constant (no adaptation). Bold lines are LOESS curve fits. Source data are provided as a Source Data file (sourcedata.xlsx). The data and code used to obtain the estimates are available at https://www.openicpsr.org/openicpsr/project/217201.

Our findings have several implications for energy and environmental policy in regions dependent on hydropower. First, the quantified health burden associated with drought-induced generation shifts provides a basis for demand-side interventions to account for this additional cost (approximately $12 billion (2019 USD) per year).

Second, our work informs ongoing policy discussions on decarbonizing electricity generation in LAC. While expanding renewable energy capacity through sources like solar and wind is essential for reducing overall emissions, this transition will not mitigate drought-related health impacts if combustion plants continue to serve as marginal generators during droughts. Moreover, the documented increase in drought exposure is expected to lead to heavier reliance on combustion power as a marginal source of energy, undermining efforts to decarbonize energy generation. Quantifying the health costs associated with this reliance may help inform cost-benefit assessments for prioritizing energy storage infrastructure alongside investments in renewable energy.

Third, the plant-level estimates provide an input for prioritizing combustion power plant retirements to alleviate the air quality

impacts of drought-induced shifts in generation. Given the projected persistence of health burdens through 2059, strategic decommissioning may be necessary even under optimistic climate scenarios. Furthermore, the evidence that these health costs disproportionately affect disadvantaged communities highlights the need to incorporate environmental justice considerations into retirement decisions. Prioritizing the decommissioning of plants in vulnerable areas can simultaneously advance climate resilience and social equity goals.

Fourth, the documented geographic variation in drought patterns also underscores the potential role of regional electricity trade in mitigating impacts on local air quality. However, without harmonized environmental regulations, interconnection risks shifting emissions to markets with weaker standards. Realizing the full benefits of interconnection would require coupling infrastructure development with coordinated policies that enforce uniform generation and environmental standards across countries.

This focus on coordinated regional solutions becomes even more critical when considering the multiple demands on water resources. While our analysis focuses on hydrological drought, environmental

flows, which help maintain river ecosystem functions, represent another important constraint on hydropower generation. This constraint is particularly relevant in Latin America, where significant alterations to river connectivity and flow regimes have been documented[48]. The complexity of these competing demands further strengthens our finding that investments in energy storage and regional grid interconnection are valuable policy tools. By reducing pressure on hydropower generation during droughts, these investments hold the potential to reduce reliance on combustion power and enable adaptive flow management that can help restore some ecosystem functions[49].

This analysis has three important caveats concerning the size of the externality and suggests that our loss estimates should be interpreted as lower bounds. First, our counterfactual calculations do not account for the disproportionate exposure to excess $PM_{2.5}$ by groups with lower socioeconomic status, whose increased vulnerability will likely lead to additional premature deaths in the region. Second, our estimates do not account for the costs created by excess $PM_{2.5}$ on other outcomes such as quality of life and productivity. Third, our estimates do not consider the cost of drought-induced excess emissions of other local or global pollutants, such as $NO_2$, $SO_2$, $O_3$, and $CO_2$, which have a wide range of environmental, health, and economic consequences. Notwithstanding the potential for even larger losses in LAC, this analysis illustrates how the existing energy generation infrastructure and droughts interact to create a considerable health burden. This burden is poised to persist and potentially worsen without energy policies that account for the water-electricity-health nexus.

## Methods

### Power plant panel
The starting point for our assembled panel is the Global Power Plant Database[50]. This database provides detailed information on the geolocation, capacity, fuel type, and commissioning year for all power plants with generators above 1 megawatt (MW). This feature of our dataset is important because little is known about the pollution burden created by plants with smaller capacities.

### Hydropower exposure to drought
Next, we augment the power plant panel with market-level measures of the degree of drought faced by hydropower plants on a monthly basis. Among various drought metrics, we focus on hydrological droughts because it allows us to directly capture the degree of water availability in hydropower watersheds. In the first step, we follow existing literature and use runoff anomalies to measure hydrological drought[14–16]. Runoff is the depth of water accumulated over time in the soil and is a valuable indicator of drought or flood conditions. Runoff anomalies indicate a period where water availability is above or below normal. Specifically, we define runoff anomalies as the difference between the monthly runoff and its corresponding average value over the reference period (2000–2019). To measure runoff anomalies, we use data from the International Energy Agency Weather for Energy Tracker database[51]. This database provides information derived from the ERA 5 reanalysis on monthly total runoff anomalies (surface and subsurface) measured in millimeters per hour (mm/h) at a spatial resolution of $0.25° \times 0.25°$.

In the second step, we determine the watershed of each hydropower plant in the sample. The watershed is the area over which water would accumulate for use by the hydropower plant. To compute these areas, we use the hydro basin polygons (areas where water collects and may flow) from the HydroSHEDS database[52]. This dataset is produced from digital elevation maps and hydrological models. Next, following standard engineering practices for each hydropower plant, we delineate the watershed by tracing all of the upstream sub-basins that flow in the direction of the power plant. The resulting dataset provides a watershed for each hydropower plant.

In the third step, we overlay the information on runoff anomalies with the watershed delineations and compute the average monthly runoff anomalies for each watershed. Supplementary Fig. 5 plots the evolution of runoff anomalies for the whole region (a) and for sub-regions defined by the IEA (b). The figure reveals that drought is not a region-wide phenomenon, with considerable sub-regional heterogeneity even during periods of significant overall drought, such as 2015–2020. To measure the impact of anomalies with a duration greater than one month, we repeat the previous calculation using a moving average of the runoff anomalies over the past three, six, nine, and 12 months. The resulting dataset provides detailed information on whether water availability conditions are above or below normal for each hydropower plant. Supplementary Fig. 6a shows the distribution of the hydropower plants in our sample and provides a visual example of how the hydrological drought dataset is constructed.

In the fourth step, we use the information derived in the previous step to compute market-level measures of the hydropower generation affected by hydrological drought. In the absence of systematic information on the boundaries of electrical markets within countries, we define each electricity market using the country boundaries. There is little cross-border trade in the region, with less than 5% of total regional generation being transmitted across countries[53].

Our preferred market-level measure is the fraction of hydropower generation capacity affected by drought (FHD). To construct this measure, we create a binary variable equal to one when the watershed of a hydropower plant has less water available than normal in the past three months (mean negative runoff anomaly) and zero otherwise. Next, we compute the average of this variable for each market and month. To account for the greater impact that larger plants experiencing drought can have on electricity generation, we weigh this average by plant capacity. We define droughts over a relatively short period (three months) because small hydropower plants are common in the region, and for these plants, even short-run changes in water availability may imply reduced generation capacity. Nonetheless, to check whether our results are robust to alternative definitions of drought duration, we also compute the FHD variable using the moving average of runoff anomalies in the past one, six, nine, and 12 months. Additionally, to verify that our results do not rely on extrapolation, we present a ridgeline plot (Supplementary Fig. 7) showing the year-by-year distribution of FHD values. The plot confirms that FHD spans the full range from 0 to 1 across all years of the study period.

We also acknowledge that different aggregation methods prioritize different features of the data. With our primary approach, we exploit the spatial granularity of our data and aim to capture the effect of both local and regional droughts by identifying periods in which hydropower generation declines due to water availability being below normal levels. However, to ensure that we study the full scope of our data, we also compute our market-level measure in other ways. For instance, we prioritize the number of plants affected in an alternative computation using an arithmetic average instead of a capacity-weighted average. Additionally, we calculate market-level measures considering only the most severe droughts. To operationalize this alternative definition, we follow Herrera-Estrada et al.[15] and create a binary variable that equals one only when runoff anomalies are one standard deviation below normal levels. We then compute the average of the variable for each market and month. These and other variations of the market-level measures help us understand the effects of extreme drought. We also compute a measure that directly gauges the intensity of the drought by calculating the average runoff anomaly for each market and month. One downside of this measure is that, in the case of non-market-wide droughts, it may incorrectly assume that negative anomalies experienced by some hydropower plants can be offset by positive anomalies experienced by other plants.

## Air quality around power plants

With our battery of market-level measures added to the panel using country and month-year identifiers, we now focus on measuring the concentration of $PM_{2.5}$ around combustion power plants. We classify plants as combustion if the primary or secondary fuel type is coal, gas, oil (including petcoke), or biomass (including waste). To measure air quality, we use information on monthly mean surface $PM_{2.5}$ concentrations measured in micrograms per cubic meter ($\mu g\,m^{-3}$) from Van Donkelaar et al.[54]. This dataset provides estimates of $PM_{2.5}$ concentrations by combining information from satellites, chemical transport models, and ground-based monitors. It provides information at $0.01° \times 0.01°$ resolution at a monthly frequency between 1998 and 2021. To integrate this information with our panel, we define each plant's dispersion area as the 50 km radius around the plant and calculate average monthly $PM_{2.5}$ concentrations for every plant. We consider this definition of the dispersion area conservative, consistent with literature indicating that $PM_{2.5}$ emissions from power plants impact areas extending at least 50 km[3,4] and significantly affect health within this distance[5–7]. While we recognize that daily exposure may vary within the dispersion area due to factors like wind direction, studies have shown that on a monthly scale, power plants in the US have similar effects on $PM_{2.5}$ levels at monitors located within 50 km, regardless of whether they are upwind or downwind[16]. These observations suggest that our monthly averages within a 50 km radius adequately measure overall exposure. Nonetheless, we also conduct analogous calculations using a 10 km radius to test result sensitivity. Supplementary Fig. 6b shows the distribution of the combustion plants in our sample and provides a visual example of how the $PM_{2.5}$ concentrations are calculated. To run placebo exercises, we also compute these measures of $PM_{2.5}$ concentrations within 50 km of non-combustion power plants (i.e., solar, wind, geothermal, and nuclear). To assess the robustness of our findings, we also construct a comparable measure of monthly $PM_{2.5}$ using data derived from ground-based monitoring stations (see Supplementary Methods 1).

## Wildfires and meteorological controls

We also include in the dataset variables that can confound the relationship between $PM_{2.5}$ concentrations and our market-level measures of drought. A particular source of concern is wildfires, which are more prevalent during droughts and lead to higher $PM_{2.5}$ concentrations[30]. To limit the influence of surface $PM_{2.5}$ related to wildfires, we follow Qiu et al.[16] and omit observations plausibly affected by wildfires. Specifically, we use data from the Global Fire Emissions Database version 4 (GFED4s)[55]. This dataset provides monthly information on carbon emissions from fires at a spatial resolution of $0.25° \times 0.25°$ and at a monthly frequency between 1997 and 2022. Using this information, we construct our primary analysis sample by excluding plant-month-year observations where fire emissions were detected within 50 km of a power plant. To evaluate the robustness of our findings, we also create two additional samples that exclude observations potentially affected by fire emissions within 75 km and 100 km. Additionally, to account for the potential impact of dust emissions, we use data from Chappell et al.[56] on seasonal sources of dust emissions (winter, spring, summer, and fall) and create three further samples that exclude plant-month observations potentially affected by either fire or dust emissions within radii of 50, 75, or 100 km.

To account for other confounding factors, we also include in our dataset several meteorological variables derived from the IEA Weather for Energy Tracker database[51,57]. These variables provide monthly frequency information and correspond to the variable mean within 50 km of the plant. The meteorological variables are temperature (°C at 2 m), total precipitation (mm/h), relative humidity (%), surface pressure (Pa), and wind speed (m/s at 10m and 100m). Additionally, to account for changes in electricity demand related to meteorological conditions (e.g., heat waves), we include market-level measures of heating degree-days (HDD) and cooling degree-days (CDD). Following IEA guidelines we define HDD (°C days) with 18 °C reference and a 15 °C threshold. Similarly CDD (°C days) is defined with 18 °C reference and a 21 °C threshold.

## Population around power plants

To quantify exposure to excess $PM_{2.5}$ concentrations, we compute the population residing within 50 km of combustion power plants. Specifically, we use $0.001° \times 0.001°$ resolution population data available between 2000 and 2020 at five-year intervals from the Global Human Settlement Layer (GHSL)[58]. To avoid double counting the population in cases where the 50 km radius for different power plants overlaps, we partition these areas using the Thiessen method. We then sum the population residing around each power plant at each time step and use linear interpolation to generate a plant-level annual time series of the population exposed.

Additionally, to better characterize the population residing around power plants, we use the Thiessen polygons derived previously and 2019 Human Development Index (HDI) data down-scaled to $0.1° \times 0.1°$ resolution[40] to compute mean plant level HDI. We then compare the plant-level HDI with country-level HDI data[59] to determine whether the population residing near power plants is systematically different.

## Benchmark statistical model

The final dataset for combustion power plants comprises 80,355 plant-month-year observations across 22 markets. However, since three markets lack hydropower capacity, the analysis dataset used for examining the impact of FHD includes 79,022 observations across 19 markets. The sample of non-combustion power plants used in the placebo exercise is constructed analogously and is comprised of 62,066 observations across 18 markets. Supplementary Table 1 presents descriptive statistics for key variables in these samples.

Using the analysis dataset, we estimate the impacts of drought on average $PM_{2.5}$ concentrations around power plants using the fixed effects regression shown in Eq. (1):

$$PM_{icmy} = \beta HD_{cmy} + \mathbf{X}'_{icmy}\boldsymbol{\gamma} + \alpha_i + \alpha_{my} + \alpha_{cm} + \varepsilon_{icmy}, \qquad (1)$$

where $PM$ represents the average $PM_{2.5}$ ($\mu g\,m^{-3}$) within 50 km of combustion power plant $i$ in electricity market $c$ in month $m$ and year $y$. $HD_{cmy}$ is the market-level measure of hydrological drought. The parameter of interest is $\beta$, which measures the impact of hydrological droughts on $PM_{2.5}$ concentrations under the assumption that the shocks are exogenous, given the controls. We account for time-invariant unobserved confounders using plant-level fixed effects ($\alpha_i$), for time-varying unobserved common shocks using month-by-year fixed effects ($\alpha_{my}$), and for market-specific seasonality using market-by-calendar-month fixed effects ($\boldsymbol{\alpha}_{cm}$). Because meteorological conditions related to hydrological droughts can also affect $PM_{2.5}$ concentrations, Eq. (1) includes a vector of meteorological controls ($\mathbf{X}'_{icmy}$). The vector comprises plant-level measures of temperature, total precipitation, relative humidity, surface pressure, wind speed measured at a height of 10 and 100 m, and an indicator of local hydrological drought. To account for the effect of fluctuations in electricity demand, potentially correlated with hydrological drought conditions, the vector also includes market-level measures of HDD and CDD.

We estimate Eq. (1) and alternative specifications using Ordinary Least Squares (OLS). To allow for arbitrary patterns of correlation among $PM_{2.5}$ concentrations across space and over time, we cluster standard errors at the market level. Reported $p$-values correspond to two-sided $t$-tests. Data analysis was performed with Stata MP v17, and figures were produced with R v4.3.2.

**Alternative statistical models and robustness checks**

Dose-response model: We use OLS to estimate Eq. (1) after discretizing the FHD variable into four groups: less than 0.25, 0.25–0.5, 0.5–0.75, and greater than 0.75. These groups roughly correspond to the quartiles of the FHD variable. In the estimation, the reference group is FHD less than 0.25.

Heterogeneous effect exercises: In the plant size exercise, as measured by capacity, we follow the US Department of Energy definition and create an indicator variable for larger than 30 MW. We then augment Eq. (1) by including an interaction term between the FHD variable and the size indicator variable. Analogously, we augment Eq. (1) in the fuel source exercise by including an interaction term with an indicator variable for the fuel source type (e.g., coal, gas, oil, or biomass). We then perform separate OLS estimations of each augmented version of Eq. (1) and compute the marginal effects, i.e., the impact of FHD on PM$_{2.5}$ for each subgroup.

Robustness checks: We validate the robustness of our findings across a diverse array of assumptions and methodological choices. Our analysis confirms that our results are not contingent on model specification. For instance, by employing the post-double selection methodology[60], we establish that our estimates remain robust even after we relax the benchmark model's assumptions of linearity and additivity, opting instead to control for higher-order polynomials of the elements of $\mathbf{X}'_{icmy}$ and their pairwise interactions. To account for confounding influences from wildfires and dust storms, we also conduct sensitivity analyses that exclude plant-month observations potentially affected by fire or dust emissions within radii of 50, 75, and 100 km. Results from these restricted samples yield very similar results, further supporting the robustness of our findings.

Additionally, we confirm the reliability of our confidence intervals by demonstrating consistent conclusions when standard errors are derived using the wild cluster bootstrap and the Conley error technique[61,62]. Furthermore, we conduct several exercises showing that we obtain similar results when we use ground-based PM$_{2.5}$ measurements and when defining the dispersion area as a 10 km radius around the power plant.

We also show that using alternative methodologies for aggregating drought measures at the market level leads to consistent results. These robustness exercises fall into two categories. First, we test alternative definitions of the FHD variable. These exercises include varying the averaging windows for hydrological droughts (1, 6, 9, and 12 months), using alternative definitions of drought occurrence (e.g., anomalies below one standard deviation), testing different weighting schemes (such as arithmetic means and weighting by generation capacity), and accounting for regional market structures by integrating neighboring countries into single markets. Across these variations, the results remain consistent, with only slight differences in point estimates.

Second, we conduct a series of exercises using the mean runoff anomaly in a market as an alternative measure of hydrological drought. Results from using this measure of drought intensity align with our benchmark findings. For example, an average drought leads to similar increases in PM$_{2.5}$ concentrations. Moreover, using this drought intensity measure alongside a spline specification allows us to test for differential effects of positive and negative runoff anomalies. We find that negative anomalies result in slightly larger increases in pollution than the reduction caused by positive anomalies, highlighting an asymmetry consistent with operational and infrastructure limitations of hydropower plants that may restrict their ability to fully utilize additional water resources. Furthermore, we leverage this measure to perform an additional placebo exercise, confirming that the relationship between PM$_{2.5}$ and hydrological drought is only observed in markets heavily reliant on hydropower. A detailed explanation of these robustness checks is provided in the Supplementary Methods 1.

**Calculation of lives lost due to excess PM$_{2.5}$**

To compute premature deaths, we combine a concentration-response function (CRF) with our estimate of excess PM$_{2.5}$ induced by drought and counts of exposed population. Our counterfactual calculation involves several steps. In the first step, we compute excess PM$_{2.5}$ by multiplying our benchmark estimate (Fig. 1a) by the observed FHD. The resulting market-by-month-year variable measures the additional air pollution observed in response to hydrological droughts shifting electricity generation to combustion power plants.

In the second step, we transform excess PM$_{2.5}$ into premature deaths using the CRF estimated by Deryugina et al.[35]. We selected this CRF for our main estimates due to its robust causal identification strategy, which leverages variation in daily wind direction to isolate the causal effect of PM$_{2.5}$ exposure on mortality. This approach minimizes the effect of potential confounders, ensuring the estimates reflect the health impacts attributable to air pollution. According to this CRF, a day of exposure to 1 µg m$^{-3}$ leads to 0.69 excess deaths per million US adults 65 or older (Medicare beneficiaries). We transform this CRF to monthly frequency by multiplying it by 30. For robustness, analogous calculations using global and regional CRFs are detailed in the Supplementary Methods 2.

In the third step, we compute the counts of the exposed population. Specifically, for each combustion power plant, we compute the count of the population 65 or older residing within 50 km by multiplying the all-age population counts by the fraction of the population 65 or older. The age distribution data[63] is available at the country-year level. Accordingly, our calculations assume a uniform age distribution within each country.

In the fourth step, we derive counts of premature deaths at the plant-month-year level by multiplying the variables described in steps one to three. Next, to account for uncertainty in our estimate of excess air pollution, in the first step, we draw our coefficient from a normal distribution with a mean equal to the estimated coefficient (1.55) and a standard deviation equal to the standard error (0.31). We then repeat 1000 times steps one to four, taking a new draw each time. In sum, we compute the number of premature deaths per plant-month-year for each draw with the following calculation: 0.69 (CRF) × 30 (days) × FHD × draw of FHD coefficient from $\sim \mathcal{N}(1.55, 0.31)$ × exposed population 65 or older. To construct Fig. 2b, we aggregate the resulting dataset to the plant level and compute the mean. To construct Fig. 2c, we aggregate the same dataset to the LAC-year level.

**Calculation of monetized lives lost due to excess PM$_{2.5}$**

To monetize the cost of the lives lost previously documented, we estimate the Value of a Statistical Life (VSL) at the country-year level in two steps. In the first step, we follow Banzhaf[64] to compute a base VSL from a meta-meta analysis of US VSL estimates. In the second step, we follow Viscusi and Masterman[65] and calculate an income-adjusted extrapolation for each country and year. We assume that the income elasticity of the VSL is equal to one and use our base VSL together with GNI per capita data[66]. Next, we compute the monetized losses in 2019 USD by multiplying our plant-month-year level dataset on premature deaths by the country and year-specific VSL estimates. To construct Fig. 2a, we aggregate the resulting dataset to the country level and compute the mean. To construct Fig. 2c, we aggregate the same dataset to the LAC-year level.

**Construction of future scenarios**

We assemble a projection dataset to study the expected exposure of hydropower plants and the cost of drought-induced excess PM$_{2.5}$. This analysis extends through 2059, aligning with the expected operational duration of existing combustion power plants. We use three scenarios constructed as combinations of the Shared Socio-economic Pathways (SSPs) and the Representative Concentration Pathways (RCPs). These SSP-RCP scenarios are SSP1–2.6, SSP2–4.5, and SSP3–7.0. The first exercise is based on monthly runoff

projections from 22 climate and earth system models from the CMPI6[67] with available information under the scenarios considered (see Supplementary Table 2). We compute runoff anomalies for each model using the projections and runoff climatologies from the IEA[51] (same as in our main analysis). We then calculate the FHD for each market using the same methodology. To assess future exposure, we use the resulting FHD series and compute, for LAC and each IEA sub-region, the percent change in mean FHD 2020–2059 relative to the mean between 2000 and 2019. This calculation results in 66 FHD projections (22 models × 3 scenarios).

To estimate premature deaths for each scenario and model, we modify the methodology used to calculate premature deaths in three ways. First, we compute excess PM$_{2.5}$ using the FHD projections instead of the observed FHD. Second, we allow for the population exposed to adjust following the SSPs demographic projections of KC and Lutz[68]. Third, when aggregating premature deaths by market and year, we incorporate the retirement of combustion power plants based on IEA forecasts[69]. Each retirement schedule is aligned with the most closely matching SSP-RCP scenario based on the expected temperature rise by 2100. Specifically, for SSP1–2.6, we use the Announced Pledges Scenario (APS), which assumes that generation from coal, gas, and oil will decline according to country pledges. For the SSP2–4.5, we use the Stated Policies Scenario (STEPS), which assumes that generation declines based on IEA's assessment of current and announced policies. For SSP3–7.0, we construct a Reference Electricity Scenario (RES), which assumes no combustion power plant retirements. All scenarios rule out the introduction of new combustion power plants and thus assume that any growth in electricity demand will be met with generation from non-combustion power plants. To present the most optimistic scenarios for PM$_{2.5}$ reduction, we also assume that plants with the largest exposed population are retired first. Our calculations also assume that combustion power plants do not benefit from technological change that would allow pollution reductions. We also assume that the concentration-response function is constant, thus ruling out that populations adopt protective measures to limit their exposure to pollution.

To estimate the monetized losses corresponding to premature deaths, we perform a calculation analogous to that performed previously but allow the country-year VSL estimates to change following the SSPs economic growth projections of Cuaresma[70].

Additionally, to isolate the contributions of each factor driving our projections (runoff, demographic and economic trends, and combustion plant retirement schedules), we systematically construct projections for each energy scenario across all SSP-RCP combinations. These results are presented in Supplementary Fig. 4.

## Reporting summary
Further information on research design is available in the Nature Portfolio Reporting Summary linked to this article.

## Data availability
The data used in this study are available in the openICPSR repository under accession code 217201[71] (https://www.openicpsr.org/openicpsr/project/217201). Source data are provided with this paper.

## Code availability
The code to reproduce the results is available in the openICPSR repository under accession code 217201.

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

## Acknowledgements
This research was supported by the World Bank Climate Support Facility Whole-of-Economy Program under TF0C1209 (M.E.). We thank A. Chappell for providing data on dust emissions sources. For valuable feedback and discussions, we thank M. Auffhammer, L. Bakkensen, J. Boomhower, D. Brewer, O. Deschenes, X. Gine, M. Kahn, N. Kala, C. Kolstad, V. Mishra, S. Sakalli, L. Taylor, C. Wichman, E. Zaveri and E. Zou. We also appreciate comments from participants at seminars and conferences at Georgia State University, Imperial College London, Georgia Institute of Technology, Javeriana University, the Ulvön conference on environmental economics, the 2nd World Bank-GWU-UVA conference on the economics of sustainable development, the OECD and the World Bank. The views expressed in this manuscript are those of the authors and do not necessarily reflect those of the World Bank or any of its affiliated organizations.

## Author contributions
M.E. and A.D.V. contributed to conceptualization, data curation, formal analysis, methodology, investigation, visualization, supervision, validation and wrote the original draft. A.D.F. carried out funding acquisition. M.E., A.D.V. and A.D.F. reviewed and edited the paper.

## Competing interests
The authors declare no competing interests.
