## [Transparent Peer Review file · Nature Communications]

Droughts Worsen Air Quality and Health by Shifting Power Generation

Corresponding Author: Professor Alejandro del Valle

Version 0:

Reviewer comments:

Reviewer #1

(Remarks to the Author)

This study examines how drought-driven reductions in hydroelectric power increase the utilization of combustion power generation, which in turn worsens air pollution. Focusing on Latin America and the Caribbean region, it analyzes air quality and meteorological data near combustion power plants to estimate drought-related changes in PM_{2.5} levels and calculate their health impacts. These findings underscore the broader implications of climate-related energy shifts, providing insights for energy and climate policies. The overall idea is very interesting, and the empirical analysis is comprehensive and convincing. However, I think the paper can strongly benefit from extra sensitivity analysis, particularly related to the choice of air quality data, pollution-mortality dose-response functions, and definitions of drought. The introduction and discussion of the paper can also be expanded to better fit the scope of Nature Communications – a general-interest journal. We hope the following comments will be helpful for your revisions.

Major:

1. One of the major concerns is related to the air quality outcomes used in this paper. The authors used modeled PM_{2.5} outcome as the main outcome variable without discussing its potential uncertainties. Given the central role of air pollution data in the entire analysis and findings, I believe more sensitivity analysis and discussions of the uncertainty are necessary. While the referenced PM_{2.5} outcome data is widely used in air pollution research, there are particular reasons to be cautious in this case because this dataset is a result of a global model, and there are not too many surface air quality monitors in the LAC area (e.g., see figure4 in <https://pubs.acs.org/doi/full/10.1021/acsestair.3c00054>).

I think the paper would be substantially stronger if the authors could implement a sensitivity analysis using the actual surface air quality observations in this region. While observational air quality data is sparse in this region, there exists data from certain countries such as Mexico and Chile. The authors could follow a similar research design in Qiu et al., 2023, where they directly linked surface air quality measurements to power plant operations. At the minimum, substantial discussion of the uncertainty of the air quality outcome data is necessary – including whether the relationship between drought and PM_{2.5} is mechanical from the chemical transport models used in Van Donkelaar et al.

2. This study uses dose-response functions of PM_{2.5}-mortality from the US-based literature as their main result. As the authors correctly pointed out, the PM_{2.5}-mortality response relationship may vary significantly by country and region. The studies the author referenced used daily pollution data covering much of the United States from 1999 to 2013, which raised issues regarding its relevance to the studied populations. I appreciate the authors making calculations based on two alternative dose-response functions (from Mexico and Chile), but it was not entirely clear why the authors didn't choose those ones for their main results. The authors did mention that they think those response functions "have the disadvantage of not being causal CRFs". Could the authors further elaborate on this point? I also suggest a sensitivity analysis based on more commonly used global response functions such as those from

3. Figure 3 and its discussion is a bit confusing as the authors combine the influences of climate change with the influences of future energy scenarios. I would suggest separating the influences of the different SSP scenarios (climate forcing) from the different energy scenarios. For example, this can be done by including two other panels in Figure 3, by showing how the results change across different SSPs (while holding the energy scenario constant), and by showing how the results change

across different energy policy scenarios (while holding the climate scenario constant).

4. I appreciate the author's effort in accounting for the wildfire influences in the analysis. However, fire emissions often have substantial influences on surface PM2.5 well beyond the range of 50km. I recommend sensitivity analysis using a higher distance radius when considering fire influences.

5. If I understand the definition of hydrological droughts correctly, it's purely based on whether the anomalies are positive or negative. Have the authors considered a sensitivity analysis that uses the "intensity" (e.g., the magnitude of the drought anomalies) for treatment definitions?

Minor:

1. In the abstract, please define more clearly "living near power plants", perhaps by including a distance metric (i.e. within 50 km?)
2. Also in the abstract, please define "hydrological drought" more precisely.
3. I think the paper will benefit from a figure and some discussion on the FHD values across different years to better help interpret the results. For example, what are the max and min values of FHD across all the years?
4. Some figures need further improvement. For instance, in Fig 2A, country abbreviations should be explained in the notes. The font size and color for the titles of similar figures are inconsistent.
5. It is unclear how the authors excluded the influences of dust on surface PM2.5, in addition to the placebo analysis. Perhaps one simple way is to drop all power plants that are close to deserts or potential dust sources.

Reviewer #2

(Remarks to the Author)

The manuscript presents a timely and important study on the relationship between hydrological droughts, air quality, and public health in Latin America and the Caribbean (LAC). The authors successfully estimate drought-induced excess PM2.5 and its health implications, a topic of growing significance in the context of climate change and energy transition. The work is highly significant for several fields, including atmospheric sciences, public health, and energy policy. By linking hydrological droughts to air pollution and mortality in LAC, the study opens new avenues for interdisciplinary research that can inform climate adaptation strategies and energy planning in the region. Given the large population exposed to these risks, this research is likely to be impactful beyond academic circles. This research significantly advances our understanding of the nexus between climate, energy, air quality, and health in Latin America and the Caribbean. It is well-executed, highly relevant, and likely to have a lasting impact on both the academic community and policymakers. I recommend its publication.

Reviewer #3

(Remarks to the Author)

Reviewer #4

(Remarks to the Author)

First of all, I would like to sincerely thank the Authors for their hard work, which resulted in the manuscript submitted for review. The presented research focuses on the impact of hydrological drought on air quality in the Latin American region. The Authors describe how the shift in energy production from traditional fossil fuels to hydropower affects air quality, expressed in PM2.5 emissions.

In general, the subject matter presented is very important and interdisciplinary. The issues between ongoing droughts, energy production, and their impact on public life are crucial in the context of the currently observed climate-economic changes on a global scale. Without a doubt, the research conducted by the Authors fills an existing gap, as there is a lack of reports in the contemporary scientific literature that describe the matters presented by the Authors.

Despite the well-prepared manuscript, some doubts arose during the reading. I believe that these should be addressed, and therefore I kindly ask the Authors to respond to the following comments:

1. Why did the Authors choose to analyze PM2.5 concentration for air quality assessment? Of course, I am aware that particles smaller than 2.5 microns can penetrate deeply into the lungs, but other fractions of particulate matter, such as PM10 or other gaseous pollutants, may also affect health. I believe that the assumption that health problems are worsened mainly by PM2.5 concentrations constitutes a major generalization. Wouldn't the impact of fires and dust emissions during atmospheric drought pose a more significant threat to air quality than shifts in energy production?
2. A weakness of the manuscript is the suggestion that the increase in PM2.5 concentration is solely a result of the shift from hydropower to combustion, while other possible sources are overlooked. I believe these should also be included in the discussion. How did the authors account for the influence of other sources of air pollution besides energy production?
3. The manuscript also suggests that the factor limiting hydropower is hydrological drought. I believe it is not the only one, as natural resources used for hydropower must also consider environmental flows, which constitute a legal barrier. How can we reconcile human energy needs, given the decreasing water resources, with the simultaneous necessity to maintain environmental flows?

4. It is not entirely clear to me why only hydrological drought was considered in the study. Hydrological drought is a consequence of persistent meteorological drought. From an air quality perspective, it is the meteorological drought that might play a key role. This drought leads to dry conditions that promote the suspension of dust and pollutants. Less rainfall means that particles, such as dust, are not washed out of the air by raindrops. Why did the authors decide to study the impact of hydrological drought instead of atmospheric drought, which might have a more direct impact on air quality?

5. The conducted research covered the Latin American and Caribbean regions. What political recommendations can be drawn from the research findings? Is it possible to introduce uniform regulations for this region, which includes many countries with different climate policies?

In general, I rate the scientific level of the presented research highly. Nevertheless, the manuscript requires revision.

Version 1:

Reviewer comments:

Reviewer #1

(Remarks to the Author)

I appreciate the author's detailed effort to address my comments. I greatly enjoyed reading the revised paper, and I think the current version is a significant improvement compared to the prior version. I do not have further comments.

Reviewer #3

(Remarks to the Author)

Reviewer #4

(Remarks to the Author)

I would like to thank the authors for the revised version of the manuscript. All my previous comments have been addressed. The authors have provided comprehensive responses to each of my suggestions. I believe the manuscript is suitable for publication in Nature Communications.

Response to the Editor and Reviewers

Manuscript Title: "Droughts Worsen Air Quality and Health by Shifting Power Generation"

Manuscript ID: NCOMMS-24-62105-A

Authors: Mathilda Eriksson, Alejandro del Valle, and Alejandro de la Fuente

Dear Editor and Reviewers,

We thank the editor and reviewers for their constructive comments and for the opportunity to revise and resubmit our manuscript to *Nature Communications*. We greatly appreciate the thoughtful feedback, which has helped us identify areas where the study can be further strengthened.

Among the main changes to strengthen the manuscript in response to reviewer feedback, we expanded our methodological analyses to include ground-based PM_{2.5} measurements, added results from alternative concentration-response functions, and extended tests accounting for wildfire and dust emissions. We also substantially developed the Introduction and the Discussion section to better articulate the contributions of the paper and to explain to readers how our findings inform public health, energy, and environmental policy. Additionally, we have reformatted the manuscript following *Nature Communications* guidelines, including proper section ordering, supplementary information formatting, and line numbering.

Below, we provide a detailed, point-by-point response to each reviewer's comment. For clarity, reviewer comments are presented in italics, followed by our responses describing how each concern has been addressed in blue. Throughout our responses, we refer to specific line numbers in the revised version of the manuscript (manuscript.pdf) and the revised version of supplementary information (supplementary_information.pdf).

To facilitate the review of all changes, we have also prepared a tracked changes version of the manuscript (manuscript_changes.pdf). Given the substantial reorganization of sections and figures in the revised manuscript, we first aligned the content of the original submission with the new structure before applying change tracking. This document also highlights new figures and those that underwent substantial revision.

Reviewer #1 (Remarks to the Author):

This study examines how drought-driven reductions in hydroelectric power increase the utilization of combustion power generation, which in turn worsens air pollution. Focusing on Latin America and the Caribbean region, it analyzes air quality and meteorological data near combustion power plants to estimate drought-related changes in PM_{2.5} levels and calculate their health impacts. These findings underscore the broader implications of climate-related

energy shifts, providing insights for energy and climate policies. The overall idea is very interesting, and the empirical analysis is comprehensive and convincing. However, I think the paper can strongly benefit from extra sensitivity analysis, particularly related to the choice of air quality data, pollution-mortality dose-response functions, and definitions of drought. The introduction and discussion of the paper can also be expanded to better fit the scope of Nature Communications – a general-interest journal. We hope the following comments will be helpful for your revisions.

Response:

We thank reviewer #1 for their thoughtful feedback and for identifying areas where the study could be further strengthened through additional sensitivity analyses. As detailed in the numbered comments below, we conducted a series of new analyses using alternative data sources and methodological approaches. These efforts allowed us to rigorously test the robustness of our findings against varying assumptions, data sources, and analytical frameworks.

To enhance the manuscript's alignment with Nature Communications' general-interest scope, we have revised the Introduction to provide a broader context, clarify the methods used, better explain the contribution of the paper to multiple strands of literature, and emphasize the policy relevance of our findings. Additionally, we expanded the Discussion section to highlight the broader policy implications of our findings.

Major:

1. One of the major concerns is related to the air quality outcomes used in this paper. The authors used modeled PM2.5 outcome as the main outcome variable without discussing its potential uncertainties. Given the central role of air pollution data in the entire analysis and findings, I believe more sensitivity analysis and discussions of the uncertainty are necessary. While the referenced PM2.5 outcome data is widely used in air pollution research, there are particular reasons to be cautious in this case because this dataset is a result of a global model, and there are not too many surface air quality monitors in the LAC area (e.g., see figure 4 in <https://pubs.acs.org/doi/full/10.1021/acsestair.3c00054>).

I think the paper would be substantially stronger if the authors could implement a sensitivity analysis using the actual surface air quality observations in this region. While observational air quality data is sparse in this region, there exists data from certain countries such as Mexico and Chile. The authors could follow a similar research design in Qiu et al., 2023, where they directly linked surface air quality measurements to power plant operations. At the minimum, substantial discussion of the uncertainty of the air quality outcome data is necessary – including whether the relationship between drought and PM2.5 is mechanical from the chemical transport models used in Van Donkelaar et al.

Response:

We thank the reviewer for raising this important concern regarding the uncertainties associated with the modeled PM2.5 data and the potential mechanical relationship between drought and

PM2.5 in chemical transport models. To address this issue, we followed the reviewer's suggestion and conducted additional sensitivity analyses using ground-based PM2.5 measurements to validate and complement the modeled PM2.5 data, which we refer to as satellite-derived PM2.5 data in the manuscript.

As the reviewer correctly points out, the availability of ground-based PM2.5 measurements in the LAC region is limited. However, we conducted an extensive search and were able to collect data from ground monitoring stations in Argentina, Brazil, Colombia, Chile, Ecuador, Peru, and Mexico, spanning 2000 to 2020. These data include hourly PM2.5 measurements from 614 stations, which we aggregated into mean monthly concentrations. For each power plant-month observation, we assigned the PM2.5 value from the nearest active monitoring station within 50 km of the plant. While this approach reduces the sample size (N=1,797), it allows a direct comparison with satellite-derived PM2.5 data.

Our analysis yielded reassuring results. The estimated effects of drought on PM2.5 using ground-based data are statistically significant and comparable in magnitude to those derived from the satellite-based dataset (Supplementary Figure 11, panel a, models 3 and 4). Interestingly, the larger effect size in model 4 likely reflects the proximity of a large fraction of ground monitors to combustion power plants (median distance = 7 km).

To further validate the satellite-derived data, we compared it with ground-based measurements. Panel c of Supplementary Figure 11 shows a strong positive correlation ($\rho = 0.82$), with some nonlinearity at higher PM2.5 levels, likely due to differences in spatial coverage: satellite-derived data represent mean concentrations within a 50 km radius, while ground monitors provide point-specific measurements. Panel d shows the distributions of PM2.5 concentrations from the two datasets, which are broadly consistent despite ground-based data exhibiting a slightly higher average.

These additional analyses strengthen the robustness of our findings and provide confidence in the reliability of satellite-derived PM2.5 data in our application. A detailed description of the analyses, including comparisons between datasets, is provided in the Supplementary Methods (lines 97-123). We have also referenced this analysis in the Methods section of the main manuscript (lines 452-454 and 549-551).

2. This study uses dose-response functions of PM2.5-mortality from the US-based literature as their main result. As the authors correctly pointed out, the PM2.5-mortality response relationship may vary significantly by country and region. The studies the author referenced used daily pollution data covering much of the United States from 1999 to 2013, which raised issues regarding its relevance to the studied populations. I appreciate the authors making calculations based on two alternative dose-response functions (from Mexico and Chile), but it was not entirely clear why the authors didn't choose those ones for their main results. The authors did mention that they think those response functions "have the disadvantage of not being causal CRFs". Could the authors further elaborate on this point? I also suggest a sensitivity analysis based on more commonly used global response functions such as those from

Response:

Thank you for raising this point about the dose-response functions (CRFs). We appreciate the opportunity to clarify our choices and expand our sensitivity analyses. Our main results rely on the CRF provided by Deryugina et al. (2019) due to its robust causal identification strategy. Their study estimates the causal effects of PM_{2.5} exposure on mortality by leveraging an instrumental variable approach using changes in local wind direction. While the CRFs from Mexico and Chile (Liu et al., 2019) provide valuable region-specific insights, their study does not prioritize causal identification. Therefore, we included them in our sensitivity analysis to complement our main results rather than using them as the basis for our primary findings. We have now extended our discussion to clarify that causal estimates are essential when using CRFs to calculate health impacts, as they ensure that the observed relationships reflect the effects of air pollution exposure rather than being confounded by other factors. Without causal identification, CRFs may capture associations influenced by variables correlated with both pollution levels and health outcomes, leading to biased estimates. We now explain our choice of preferred CRF to readers in the Methods section (lines 581-589).

We agree with the referee that extending the analysis provides readers with a clearer understanding of how our results compare under alternative CRF estimates. While we noticed that a citation at the end of the comment may have been unintentionally omitted, we sought to follow the spirit of the suggestion and expand the sensitivity analysis to include three widely recognized CRFs (including those used by the Global Exposure Mortality Model, the US EPA, and the WHO). Supplementary Figure 1 presents the results for premature deaths, and Supplementary Figure 3 shows the corresponding monetized losses across all six CRFs considered. This expanded analysis demonstrates that our preferred CRF provides mortality estimates in the mid-range of the various CRFs. We have also included a new sub-section in the Supplementary Methods, "Alternative Calculation of Lives Lost and Losses Due to Excess PM_{2.5}," which details the methodology for calculating premature deaths and losses using each of the alternative CRFs (Supplementary Methods, lines 191-230). The main text has also been updated, revising the section previously discussing the Liu et al. (2019) sensitivity test to summarize the conclusions from this extended analysis (Results, lines 191-194 and 213-214).

3. Figure 3 and its discussion is a bit confusing as the authors combine the influences of climate change with the influences of future energy scenarios. I would suggest separating the influences of the different SSP scenarios (climate forcing) from the different energy scenarios. For example, this can be done by including two other panels in Figure 3, by showing how the results change across different SSPs (while holding the energy scenario constant), and by showing how the results change across different energy policy scenarios (while holding the climate scenario constant).

Response:

Thank you for your suggestion to disentangle the effects of SSP-RCPs and energy policy scenarios. We agree that clarifying the roles of these different components improves the

interpretability of our projections. Each projection in our analysis combines three different components: runoff projections from SSP-RCP scenarios to estimate future drought, socioeconomic projections from SSPs to assess demographic factors and economic growth, and energy policy scenarios to model the retirement of combustion power plants. To provide a clearer explanation of how these components are integrated into the future pathways, we have revised the Methods section (lines 637-638 and 654-657) and the Results section (lines 220-229 and 239-246).

We appreciate the referee's suggestion to include a figure that separates the influences of these components. As the main text figure is already composed of 8 subfigures, we have chosen to create a new supplementary figure containing six subfigures (Supplementary Figure 4). The first row of sub-figures depicts premature deaths, and the second row displays monetized losses. Each column represents one energy policy scenario (APS, STEPS, and RES), and within each subfigure, variations across SSP-RCP scenarios are illustrated. This layout effectively separates the influence of energy policy scenarios from SSP-RCP scenarios and highlights that energy policy is the primary driver of the differences between projections.

Our analysis, holding energy policy constant, shows that SSP-RCP rankings are influenced by climate, demographic, and economic factors. While higher climate forcing (SSP-RCP) scenarios predict increased droughts as expected, differences in runoff projections by 2059 are relatively small (as shown in Figure 3, panel a). This indicates that demographic factors play a more significant role. For instance, SSP1-2.6 leads to the largest increase in premature deaths, driven by the faster aging of the population, which offsets minor differences in runoff across SSPs. Additionally, higher economic growth in SSP1-2.6 results in greater monetized losses due to increases in the value of a statistical life (VSL). We have updated the results section (lines 257–270) to explain these findings to readers. Our revisions emphasize that the future trajectory of drought-induced premature deaths will primarily depend on energy policy. In particular, the retirement schedule of combustion power plants.

4. I appreciate the author's effort in accounting for the wildfire influences in the analysis. However, fire emissions often have substantial influences on surface PM2.5 well beyond the range of 50km. I recommend sensitivity analysis using a higher distance radius when considering fire influences.

Response:

We thank the referee for the suggestion to consider a larger distance radius to account for the influence of fire emissions on surface PM2.5. Following your suggestion, we conducted a sensitivity analysis extending the exclusion radius from 50 km to 75 km and 100 km, as shown in Supplementary Figure 9. Models 1, 3, and 5 exclude observations potentially affected by fire emissions within 50, 75, and 100 km, respectively. Additionally, to address minor comment 5 about dust emissions (discussed later), models 2, 4, and 6 extend this exclusion to observations potentially influenced by fire or dust emissions within the same radii.

The results confirm the robustness of our main findings across different exclusion thresholds. Although larger exclusion radii yield slightly wider confidence intervals, given the reduced sample size, the point estimates indicate similar effect sizes in all cases. We have included a discussion of these findings in the Supplementary Methods (lines 39-54) and revised the main text to reference this sensitivity analysis in the Results section (lines 135-139). We have also revised the Methods section (lines 465–471, 542-546) to explain the construction of these samples.

5. If I understand the definition of hydrological droughts correctly, it's purely based on whether the anomalies are positive or negative. Have the authors considered a sensitivity analysis that uses the "intensity" (e.g., the magnitude of the drought anomalies) for treatment definitions?

Response:

We agree that incorporating the intensity of drought anomalies as a treatment variable is an important sensitivity check. While the robustness check section in the Supplementary Information contained these exercises (Supplementary Figure 14 in the revised manuscript, and "Alternative statistical models and robustness checks", lines 158–190), we recognize that we did not reference or sufficiently describe them in the main text. We thank the reviewer for bringing this issue to our attention.

To briefly summarize our exercises in this area, we use the mean runoff anomaly among hydropower plants in a market as an alternative measure of hydrological drought. This measure is standardized into units of standard deviation for easier interpretation and replaces the FHD variable in Equation 1. The results, shown in Supplementary Figure 14 (model 1), align closely with our benchmark findings; for instance, the implied PM2.5 concentration for an average-intensity drought is 16.86 mug/m³, comparable to the 16.59 mug/m³ implied by our benchmark using the mean FHD.

This intensity measure also enables additional analyses. Using a spline specification with a kink at zero, we test whether the effects of negative runoff anomalies (droughts) on pollution differ from those of positive runoff anomalies (downpours). The results (model 2) show that droughts increase pollution slightly more than downpours reduce it. This is consistent with the operational and infrastructure limitations of hydropower plants, which may prevent them from fully utilizing additional hydrological resources once capacity is reached. Moreover, the mean runoff anomaly can be computed for all power plants, not just hydropower plants, allowing us to analyze markets with no or limited hydropower capacity. This adjustment enables an additional placebo test, confirming that the relationship between PM2.5 and hydrological drought is specific to markets reliant on hydropower. The results of this analysis are presented in models 3 to 5.

Overall, while this intensity measure provides valuable supporting evidence and further strengthens the case for the proposed mechanism, we continue to prefer the FHD as our primary measure. A key limitation of the intensity measure is that, in cases of non-market-wide

droughts, it may incorrectly assume that negative anomalies at some hydropower plants can be offset by positive anomalies at others.

To provide readers with a clearer understanding of these results, we have detailed the construction of this hydrological drought measure and our rationale for preferring the FHD in the Methods section (lines 424–428). Additionally, the main text now includes a summary of this analysis (lines 561–573) and references the more comprehensive discussion and results available in the Supplementary Information (lines 158–190). We have also revised the discussion in Supplementary Information for clarity.

Minor:

1. In the abstract, please define more clearly “living near power plants”, perhaps by including a distance metric (i.e. within 50 km?)

Response:

We agree with this suggestion and have revised the abstract to provide a more precise definition. Specifically, we replaced "live near a combustion power plant" with "live within 50 km of a combustion power plant."

2. Also in the abstract, please define “hydrological drought” more precisely.

Response:

We also agree that providing a more precise definition of "hydrological drought" enhances clarity. In response, we have revised the abstract to explicitly define hydrological droughts as "negative runoff anomalies in hydropower watersheds." We have also taken the opportunity to edit the abstract to improve overall clarity.

3. I think the paper will benefit from a figure and some discussion on the FHD values across different years to better help interpret the results. For example, what are the max and min values of FHD across all the years?

Response:

We appreciate the suggestion to include a figure and discussion on the FHD values across different years to aid in interpreting the results. We have followed the recommendation and added Supplementary Figure 7, which presents the annual distribution of FHD values over the study period (2000–2020). The ridgeline plot shows substantial inter-annual variability and confirms that FHD values are well-represented across their entire range (0 to 1) in all years. We have also updated the Methods section to refer to this figure and explicitly note that our results do not rely on extrapolation, as the FHD support extends across its full range each year (Methods, lines 408–411).

Additionally, we have included a summary statistics table (Supplementary Table 1) to provide further context for the data used in the analysis. This table reports the mean, standard deviation, min, and max for FHD and other key variables in both our analysis dataset and the placebo exercise dataset.

4. Some figures need further improvement. For instance, in Fig 2A, country abbreviations should be explained in the notes. The font size and color for the titles of similar figures are inconsistent.

Response:

We appreciate the reviewer's suggestions regarding figure improvements. In Figure 2, we have implemented several improvements to enhance clarity and consistency. For panel a, we added all country abbreviations used in the figure note. In panel b, we recreated the map natively in R, allowing better control over font sizes to ensure consistency with other panels. Additionally, in panel c, we included a legend to clearly indicate the series and updated the axis titles to consistent black coloring for improved readability.

Beyond these specific revisions to Figure 2, we also took the opportunity to refine the formatting and layout of all figures throughout the manuscript to enhance their overall presentation and coherence.

5. It is unclear how the authors excluded the influences of dust on surface PM2.5, in addition to the placebo analysis. Perhaps one simple way is to drop all power plants that are close to deserts or potential dust sources.

Response:

This is a very helpful suggestion. To operationalize the definition of dust sources, we took advantage of the dataset produced by Chappell et al. (2023), which identifies seasonal (winter, spring, summer, and fall) dust emission sources for the world. We then use this information to identify plant-month observations potentially affected by dust emissions within 50, 75, or 100 km.

Since this comment closely relates to your major comment 4 regarding the exclusion of observations potentially affected by fires and extending the radii, we combined these analyses. Specifically, we conducted a sensitivity analysis excluding observations potentially influenced by fire or dust emissions within radii of 50, 75, and 100 km. The results of this extended analysis are shown in Supplementary Figure 9 (models 2, 4, and 6). As noted in our response to comment 4, the findings are reassuringly robust and demonstrate that even when excluding observations potentially affected by fires or dust emissions, we observe very similar results. We have added a description of this sensitivity analysis in the Supplementary Methods (lines 39-54) and updated the main text to reference these additional tests in the Results section (lines

135-139). We have also revised the Methods section (lines 465–471, 542-546) to explain how we constructed the samples and the data used to identify dust emission sources.

Reviewer #2 (Remarks to the Author):

The manuscript presents a timely and important study on the relationship between hydrological droughts, air quality, and public health in Latin America and the Caribbean (LAC). The authors successfully estimate drought-induced excess PM_{2.5} and its health implications, a topic of growing significance in the context of climate change and energy transition.

The work is highly significant for several fields, including atmospheric sciences, public health, and energy policy. By linking hydrological droughts to air pollution and mortality in LAC, the study opens new avenues for interdisciplinary research that can inform climate adaptation strategies and energy planning in the region. Given the large population exposed to these risks, this research is likely to be impactful beyond academic circles.

This research significantly advances our understanding of the nexus between climate, energy, air quality, and health in Latin America and the Caribbean. It is well-executed, highly relevant, and likely to have a lasting impact on both the academic community and policymakers. I recommend its publication.

Response:

We thank Reviewer #2 for the encouraging feedback on our manuscript. We are pleased that you found the study relevant and impactful across multiple fields, and we appreciate your recognition of its contribution to understanding the nexus between climate, energy, air quality, and health in Latin America and the Caribbean. Thank you for your thoughtful evaluation and recommendation for publication.

Reviewer #3 (Remarks to the Author):

Response:

We thank Reviewer #3 for their contribution to the review process and appreciate the Nature Communications initiative to support Early Career Researchers in peer review. This collaborative approach not only provides valuable training for researchers at the early stages of their careers but also enriches the peer review process. We appreciate the constructive feedback provided through this joint effort.

Reviewer #4 (Remarks to the Author):

First of all, I would like to sincerely thank the Authors for their hard work, which resulted in the manuscript submitted for review. The presented research focuses on the impact of hydrological drought on air quality in the Latin American region. The Authors describe how the shift in energy production from traditional fossil fuels to hydropower affects air quality, expressed in PM2.5 emissions.

In general, the subject matter presented is very important and interdisciplinary. The issues between ongoing droughts, energy production, and their impact on public life are crucial in the context of the currently observed climate-economic changes on a global scale. Without a doubt, the research conducted by the Authors fills an existing gap, as there is a lack of reports in the contemporary scientific literature that describe the matters presented by the Authors.

Despite the well-prepared manuscript, some doubts arose during the reading. I believe that these should be addressed, and therefore I kindly ask the Authors to respond to the following comments:

Response:

We thank Reviewer #4 for their thoughtful remarks on our manuscript. We are pleased that you found the study to address an important and interdisciplinary topic, as well as its contribution to filling a gap in the existing literature. We are also grateful for the opportunity to improve the manuscript based on your feedback. Below, we address each of your comments.

1. Why did the Authors choose to analyze PM2.5 concentration for air quality assessment? Of course, I am aware that particles smaller than 2.5 microns can penetrate deeply into the lungs, but other fractions of particulate matter, such as PM10 or other gaseous pollutants, may also affect health. I believe that the assumption that health problems are worsened mainly by PM2.5 concentrations constitutes a major generalization. Wouldn't the impact of fires and dust emissions during atmospheric drought pose a more significant threat to air quality than shifts in energy production?

Response:

We thank the reviewer for their thoughtful comment regarding our focus on PM2.5 for assessing air quality impacts. This is an important point, and we agree that other pollutants, such as PM10, NO₂, SO₂, and O₃, may also be produced by operating combustion power plants and significantly affect air quality and health outcomes. We also acknowledge that we did not sufficiently explain the rationale behind our focus on PM2.5, and we are grateful to the reviewer for bringing this issue to our attention.

Our decision to focus on PM2.5 was guided by several considerations, including the robust literature documenting the health impacts of PM2.5, data availability constraints, the

methodological challenges of disentangling the effect of strongly correlated pollutants, and the overall objectives of the study.

First, from a health perspective, exposure to PM_{2.5} is important to study because it disperses over wide areas and because there is robust epidemiological evidence linking it to adverse health outcomes, particularly mortality. We have revised the introduction to clarify our motivation to focus on PM_{2.5} to readers early on (lines 9-15). Although other pollutants are also important contributors to mortality and morbidity risk, we think that PM_{2.5} provides a useful starting point and a robust lower bound of the health impact of drought-induced worsening of air quality. We have also acknowledged that our focus on PM_{2.5} is a limitation and have clarified in the Discussion section (lines 343–345) that our estimates likely underestimate the total health burden because they do not account for the effect of pollutants such as NO₂, O₃, and SO₂.

Another related and important consideration is that focusing on PM_{2.5} allows us to leverage well-established concentration-response functions (CRFs), which are widely used by government agencies to quantify health impacts and guide policymaking. While CRFs for other pollutants exist, their use in policy contexts is less common. By grounding our analysis in methodologies that are well-established and already accepted by policymakers, we ensure that our results are both scientifically robust and actionable. As previously mentioned in response to referee 1 point 2, we have extended our counterfactual calculations of premature deaths to include computations based on widely used PM_{2.5} CRFs, including those used by the WHO, the EPA, and GEMM. Please see Figure S1 and the alternative calculation of lives lost and losses due to excess PM_{2.5} in the supplementary information (lines 191-230).

Second, we aimed to assess air quality impacts across the entire Latin America and Caribbean region rather than confining our analysis to a few countries with extensive ground-based monitoring networks. This broader geographical scope required air quality data with both high spatial and temporal resolution spanning two decades. Recent advances in satellite-derived PM_{2.5} measurements enabled us to construct a comprehensive monthly plant-level panel dataset covering 2000-2020. This dataset provided the necessary foundation for conducting a rigorous analysis of air quality impacts at a regional scale. Unfortunately, to our knowledge, datasets for other pollutants lack similar attributes. For instance, available data on PM₁₀ and surface-level O₃ typically has a more limited spatial resolution and inconsistent temporal coverage compared to PM_{2.5}. NO₂ data is available in high spatial and temporal resolution for recent years (e.g., from TROPOMI), but the series only goes back to 2017, making it unsuitable for analyzing long-term trends. While NO₂ data from OMI offers a longer time series, its lower spatial resolution limits its utility for our application. Accordingly, we consider PM_{2.5} to be the most appropriate pollutant for this study, partly due to the availability of comprehensive, high-quality data that enables a robust long-term regional analysis. In Methods (lines 433-438), we explain to readers why the PM_{2.5} data is ideally suited for our analysis.

The third reason is that analyzing multiple pollutants simultaneously presents a considerable methodological challenge. For example, PM_{2.5}, PM₁₀, NO₂, and O₃ are often co-emitted and highly correlated, making it difficult to isolate their individual effects. In addition, NO₂, O₃, and

SO₂ have distinct sources, dispersion patterns, and interactions with meteorological conditions, which would require us to use a different methodological approach for estimating their impact and aggregating their damages. Our focus on PM_{2.5} makes the analysis more tractable and allows us to isolate and quantify the impact of droughts on a key pollutant through its effects on the energy system.

Regarding the reviewer's point about the importance of fires and dust emissions during droughts, we agree that these are very large contributors to air quality degradation. Indeed, recent research in the United States has shown that wildfires have offset the gains achieved through air quality regulations. Similarly, dust storms are known to have considerable health impacts. Nonetheless, our study is distinct not because of the magnitude of the effect but rather because of its focus on an additional novel mechanism, namely, that hydrological droughts alter air quality through its effect on the energy system. Importantly, this mechanism operates independently of fires and dust storms and must be addressed by different policy interventions. For instance, addressing energy system-related air quality impacts may involve diversifying the energy mix, improving the interconnection between electricity markets, or investing in alternative marginal sources of energy such as batteries. By quantifying the cost of drought-induced excess PM_{2.5} through the energy system, we provide valuable input for policymakers by demonstrating that this externality is substantial enough to be considered a significant problem and by providing plant-level measures of its cost for use in cost-benefit analyses of energy investments. We have revised the Introduction (lines 79-94) and the Discussion section (lines 294-327) to better articulate our contribution and the policy implications of our findings. Further details on the revisions to the paper's policy discussion can be found in our response to comment 5 below.

2. A weakness of the manuscript is the suggestion that the increase in PM_{2.5} concentration is solely a result of the shift from hydropower to combustion, while other possible sources are overlooked. I believe these should also be included in the discussion. How did the authors account for the influence of other sources of air pollution besides energy production?

Response:

We appreciate the referee's comment highlighting the potential influence of non-energy-related sources of PM_{2.5}. While we recognize that several significant sources of PM_{2.5} exist beyond energy production, these sources would need to be correlated with both drought conditions and PM_{2.5} concentrations to confound our analysis. To address this concern, our study employs multiple measures to account for the effects of wildfires and dust storms, which are associated with droughts and could potentially confound our estimates. Specifically, we provide robust evidence demonstrating both that these alternative sources are unlikely to drive our findings and that the observed pattern of results is best explained by shifts in energy generation to combustion power plants. Below, we outline the steps taken to address this issue.

First, we designed our dataset to closely align with the hypothesis being tested, ensuring that our measures of air quality and drought impacts on the energy system directly reflect the proposed mechanism. Specifically, PM_{2.5} concentrations are measured within a 50 km radius of

combustion power plants, where emissions from energy production are most likely to influence air quality. For hydrological drought, we utilize runoff anomaly data measured in hydropower watersheds, capturing the direct link between drought severity and disruptions in hydropower availability. By focusing on these proximate measures, we increase the likelihood that observed changes in PM_{2.5} concentrations are tied to drought-induced energy system shifts rather than unrelated phenomena.

Second, our empirical strategy incorporates a comprehensive set of controls and fixed effects to mitigate the influence of confounders. In estimating Equation (1), we control for meteorological factors such as temperature and precipitation, as well as proxies for energy demand. Additionally, we include fixed effects to account for time-invariant unobserved factors (e.g., geographic characteristics or persistent emissions from industry and transportation), unobserved time-varying shocks common to the regions (e.g., economic trends or policy changes), and market-specific seasonal patterns (e.g., wildfire season or seasonal fluctuations in other sources of emissions). Together, these controls and fixed effects help isolate the effect of hydrological droughts on PM_{2.5} concentrations by reducing the risk of bias from non-energy-related sources, ensuring that our estimates are more directly attributable to the hypothesized mechanism of drought-induced shifts in energy production.

Third, we take specific steps to address the confounding effects of wildfires by excluding plant-month observations where fire emissions are detected within a 50 km radius of combustion power plants, following the methodological recommendations of Qiu et al. (2023). In this revision, we have extended this analysis by expanding the exclusion radius to 75 km and 100 km and jointly excluding observations potentially affected by both fires and dust emissions (Supplementary Figure 9 and discussion in Supplementary Methods, lines 39-54). Across these specifications, the results remain consistent, with minimal variation in point estimates, suggesting that wildfire and dust emissions are unlikely to drive our findings.

Nonetheless, emissions from fires or dust storms can travel hundreds of kilometers, and the previous analyses cannot fully rule out their confounding impact. To address this, we conducted two placebo exercises. The first placebo test reproduces the previous analysis but uses PM_{2.5} concentrations around non-combustion power plants (e.g., wind, solar, geothermal, nuclear) as the outcome. If wildfires or dust storms were the primary drivers of increased PM_{2.5} during droughts, we would expect to observe similar effects around non-combustion power plants. However, our results show no statistically significant effect of drought on PM_{2.5} in these areas, supporting the conclusion that the observed increases are linked to combustion power plants. A caveat of this exercise is that combustion and non-combustion plants may be located in systematically different locations, and the null result could partly reflect differences in geographic characteristics. To address this limitation, we conduct a second placebo test. The second test evaluates PM_{2.5} concentrations around combustion power plants in the years before they became operational. This exercise allowed us to test whether drought-induced increases in PM_{2.5} might be explained by confounding factors unrelated to plant operations, such as regional wildfire or dust activity. Once again, the results showed no significant effect,

ruling out these alternative explanations. Together, these placebo tests confirm that our findings are unlikely to be driven by wildfires or dust storms.

Finally, our analysis of heterogeneous effects provides strong support for the energy system mechanism. We find that PM_{2.5} increases are largest for smaller power plants, which tend to be air-cooled, and those that use biomass and oil as fuels. These plants often serve as backup or peaking generation units and are more likely to ramp up generation during droughts when hydropower availability is constrained. In contrast, larger coal and gas plants, which generally operate near maximum capacity and provide baseload power, exhibit smaller adjustments during droughts. This pattern aligns closely with the hypothesis that droughts increase air pollution by shifting generation from hydropower to combustion power plants with spare capacity. We view these multiple lines of evidence, robust controls, extended exclusion analyses, placebo tests, and heterogeneity findings as collectively supporting the conclusion that the estimated relationship between hydrological droughts and excess PM_{2.5} concentrations is primarily driven by changes in energy production rather than other drought-related phenomena such as wildfires or dust storms.

We acknowledge that the original manuscript did not articulate our strategy to isolate the mechanism clearly, and we thank the referee for encouraging us to provide a clearer explanation of this issue. We have revised the manuscript to provide a brief discussion of the ideas presented above and specifically to clarify to readers how these multiple lines of evidence indicate that the most likely mechanism for the estimated increase in PM_{2.5} is the shift in generation to combustion power plants. Additionally, we have taken the opportunity to explain that the mechanism section titled “Shifts to combustion power explain excess PM_{2.5}” has a more narrow scope as it does not aim to quantify the impact of droughts on PM_{2.5} through alternative mechanisms (e.g., increases in PM_{2.5} due to wildfires) but focuses only on providing supporting evidence to explain the factors driving the increase in PM_{2.5} estimated in the previous sub-section “Droughts affecting hydropower lead to excess PM_{2.5}”. Please see the revised Introduction (lines 16-53), Results (lines 130-179), and Methods(509-518 and 525-536).

3. The manuscript also suggests that the factor limiting hydropower is hydrological drought. I believe it is not the only one, as natural resources used for hydropower must also consider environmental flows, which constitute a legal barrier. How can we reconcile human energy needs, given the decreasing water resources, with the simultaneous necessity to maintain environmental flows?

Response:

We thank the reviewer for raising this point about the role of environmental flows in hydropower generation. We fully agree that hydrological drought is not the only factor limiting hydropower capacity. Environmental flows are essential for sustaining downstream ecosystems and maintaining long-term ecological balance, creating additional constraints on water availability for energy production, particularly during periods of drought.

While the primary focus of our study is quantifying a previously unmeasured externality, the public health costs of drought-induced shifts to combustion power, we recognize the importance of situating these findings within the broader context of competing water demands. This is particularly relevant in Latin America, where significant alterations to river connectivity and flow regimes have been documented (Grill et al. 2019). In response to this comment, we have revised the Discussion section (lines 327-332) to explicitly acknowledge that environmental flow requirements, alongside hydrological drought, play a critical role in shaping hydropower capacity.

Moreover, our analysis also suggests that the infrastructure investments needed to address the health costs we document could have important synergies with environmental flow management. The revised Discussion section (lines 332-336) also highlights how, for instance, investments in energy storage and regional grid interconnection can simultaneously reduce reliance on combustion power during droughts while enabling adaptive flow management that can help restore some ecosystem functions (Poff et al. 2016). We have also taken the opportunity to add the Grill et al. (2019) and (Poff et al. 2016) references to the paper.

4. It is not entirely clear to me why only hydrological drought was considered in the study. Hydrological drought is a consequence of persistent meteorological drought. From an air quality perspective, it is the meteorological drought that might play a key role. This drought leads to dry conditions that promote the suspension of dust and pollutants. Less rainfall means that particles, such as dust, are not washed out of the air by raindrops. Why did the authors decide to study the impact of hydrological drought instead of atmospheric drought, which might have a more direct impact on air quality?

Response:

As the reviewer correctly points out, the direct relationship between meteorological drought and PM2.5 concentrations is important to recognize. Our focus on hydrological drought is intentional and aligns with both the primary aim of our study and recent literature examining drought impacts on electricity systems in the western US (Eyer and Wichman 2018; Herrera-Estrada et al. 2018; Qiu et al. 2023). Specifically, we aim to isolate and quantify how drought impacts air quality through its influence on energy systems. Hydrological drought, measured through runoff anomalies in hydropower watersheds, offers a direct metric of water availability for hydropower generation. This measure allows us to trace a clear causal chain from reduced water availability to our measure of the impact of drought on affected hydropower generation (e.g., FHD) and ultimately to higher PM2.5 concentrations. We now better motivate this methodological choice early on to readers in the Introduction (lines 19-24) and later on in the Methods section (lines 358-390).

We agree with the referee and fully acknowledge that meteorological drought has direct effects on air quality, such as reduced particle washout by precipitation and increased dust suspension. To account for these effects, our empirical strategy includes an extensive set of meteorological controls, including temperature, precipitation, relative humidity, wind speed, and surface pressure (detailed in Methods, lines 472-481 and 512-518). To further account for potential

non-linear relationships and interactions between these variables, we employ more flexible specifications in our robustness checks. Specifically, we allow for quadratic and cubic polynomials of these controls as well as all their pairwise interactions. Given the large number of potential controls this introduces, we employ post-double selection LASSO methodology to identify the most flexible and relevant specifications (detailed in Methods, lines 537-542 and Supplementary Methods, lines 23-38). Our results remain robust across these specifications, suggesting that the observed effects are indeed driven by energy system responses to hydrological drought rather than direct meteorological impacts. We also now mention in the introduction that we flexibly control for these meteorological factors (lines 28-34). We thank the reviewer for prompting us to better articulate these ideas and to better explain this important aspect of our research design.

5. The conducted research covered the Latin American and Caribbean regions. What political recommendations can be drawn from the research findings? Is it possible to introduce uniform regulations for this region, which includes many countries with different climate policies?

Response:

We thank the reviewer for their thoughtful feedback and for highlighting the importance of addressing the broader policy implications of our findings. In response, we have rewritten the Discussion section to expand the analysis of how our results inform energy and environmental policy in LAC and beyond. The revised Discussion explores key policy opportunities, focusing on integrated water-energy planning, regional collaboration, and environmental justice.

Although uniform regulations across the region may be challenging given the diversity of political, economic, and climatic conditions, the Discussion emphasizes the potential for regional collaboration. For instance, strengthening transnational electricity grids could mitigate the need for local combustion power plants during droughts by allowing countries with surplus hydropower capacity to export electricity to regions experiencing shortages. Since our analysis shows that droughts do not typically affect all regions simultaneously, interconnected grids can exploit temporal and spatial variability in drought conditions. However, the revised Discussion also stresses that coordination is essential to harmonize generation and environmental standards across interconnected regions, ensuring that emissions are not disproportionately shifted to areas with weaker regulations.

The revised Discussion also identifies policies that countries can implement independently to address the substantial health costs quantified in our study. We highlight demand-side management as a practical measure to reduce peak loads and reliance on combustion plants during droughts.

In the context of regional decarbonization, the revised Discussion also elaborates on the need to complement renewable energy expansion with investments in energy storage technologies. Our findings emphasize that renewable energy alone will not mitigate drought-related health impacts if combustion power plants remain the marginal generators during droughts. The health costs quantified in our study provide valuable input for cost-benefit analyses of energy storage

infrastructure, demonstrating their potential to deliver significant public health benefits alongside environmental gains.

Furthermore, the Discussion expands on how our findings can guide the prioritization of combustion power plant retirements. Our plant-level estimates of population exposure highlight the critical need for strategic decommissioning, particularly given the projected persistence of health burdens beyond mid-century, even under optimistic climate-forcing scenarios. Notably, our observation that these health costs disproportionately affect disadvantaged communities underscores the importance of incorporating environmental justice into policymaking. Targeted strategies, such as prioritizing the retirement of combustion power plants in densely populated and socioeconomically vulnerable areas, hold significant potential to simultaneously address health disparities and reduce economic and environmental inequalities.

The revised Discussion section (lines 294-326) aims to provide a nuanced yet actionable analysis of our research implications, informing both regional coordination and country-specific policymaking. We hope these additions address your feedback and strengthen the relevance of our findings for policymaking.

In general, I rate the scientific level of the presented research highly. Nevertheless, the manuscript requires revision.

Response:

Thank you for your careful reading and helpful comments, which greatly improved the paper.